**Indian winter and summer monsoon strength over the 4.2 ka BP event in foraminifer**
**isotope records from the Indus River delta in the Arabian Sea**
Alena Giesche[1], Michael Staubwasser[2], Cameron A. Petrie[3], and David A. Hodell[1]
*[1] Godwin Laboratory for Palaeoclimate Research, Department of Earth Sciences, University of*
*Cambridge, Cambridge, CB2 3EQ, United Kingdom*
*[2] Institute for Geology und Mineralogy, University of Cologne, Zülpicher Str. 49a, 50674 Cologne,*
*Germany*
*[3] Department of Archaeology, University of Cambridge, Cambridge, CB2 3DZ, United Kingdom*
*Correspondence to:* Alena Giesche (ag927@cam.ac.uk)

**Abstract**

The plains of northwest South Asia receive rainfall during both the Indian Summer (June-September) and Winter (December-March) Monsoon. Researchers have long attempted to deconstruct the influence of these precipitation regimes in paleoclimate records, in order to better understand regional climatic drivers and their potential impact on human populations. The Mid-Late Holocene transition between 5.3-3.3 ka BP is of particular interest in this region because it spans the period of the Indus Civilization from its early development, through its urbanization and onto eventual transformation into a rural society. An oxygen isotope record of the surface-dwelling planktonic foraminifer *Globigerinoides ruber* from the northeast Arabian Sea provided evidence for an abrupt decrease in rainfall and reduction in Indus River discharge at 4.2 ka BP, which the authors linked to the decline of the urban phase of the Indus Civilization (Staubwasser et al., 2003). Given the importance of this study, we used the same core (63KA) to measure the oxygen isotope profiles of two other foraminifer species at decadal resolution over the interval from 5.4 to 3.0 ka BP, and replicate a larger size fraction of *G. ruber* than measured previously. By selecting both thermocline-dwelling (*Neogloboquadrina dutertrei*) and shallow-dwelling (*Globigerinoides sacculifer*) species, we provide enhanced detail of the climatic changes that occurred over this crucial time interval. We found evidence for a period of increased surface water mixing, which we suggest was related to a strengthened winter monsoon with a peak intensity over 200 years from 4.5 to 4.3 ka BP. The time of greatest change occurred at 4.1 ka BP when both the summer and winter monsoon weakened, resulting in a reduction in rainfall in the Indus region. The earliest phase of the urban Mature Harappan period coincided with the period of inferred stronger winter monsoon between 4.5-4.3 ka BP, whereas the end of the urbanized phase occurred some time after the decrease in both the summer and winter monsoon strength by 4.1 ka BP. Our findings provide evidence that the initial growth of large Indus urban centers coincided with increased winter rainfall, whereas the contraction of urbanism and change in subsistence strategies followed a reduction in rainfall of both seasons.

## 1. Introduction

The ~4.2 ka BP event is considered to be a defining event of the Mid-Late Holocene transition period (Mayewski et al., 2004), and is marked by intense aridity in much of western Asia, which has been linked to cultural transitions in Mesopotamia, Egypt, and the Indus Civilization (Staubwasser and Weiss, 2006; Weiss, 2016). Recently, a climate reconstruction from Mawmluh cave in northeastern India has been used to formally demarcate the post-4.2 ka BP time as the Meghalayan Age (Letter from the 44[th] International Union of Geological Sciences, 2018; Walker et al., 2012). However, defining the exact timing and extent of aridity at ~4.2 ka BP remains an open question (Finné et al., 2011; Wanner et al., 2008). In this special issue devoted to the "4.2 ka event", we provide new paleoclimate data from a marine core in the northern Arabian Sea over this critical time interval to better understand the changes that occurred in both winter and summer hydroclimate over the Indian Subcontinent.

The $\delta^{18}O$ record of *Globigerinoides ruber* from marine core 63KA, obtained from the Arabian Sea off the coast of Pakistan and produced by Staubwasser et al. (2003), was among the first well-resolved paleoclimate records to suggest a link between a decrease in Indus River discharge around 4.2 ka BP and the decline of the urban phase of the Indus Civilization. Since the publication of this record, several other terrestrial paleoclimate reconstructions from the region (Berkelhammer et al., 2012; Dixit et al., 2014, 2018; Giosan et al., 2012; Kathayat et al., 2017; Menzel et al., 2014; Nakamura et al., 2016; Prasad and Enzel, 2006), and a number of marine reconstructions (Giosan et al., 2018; Gupta et al., 2003; Ponton et al., 2012) have added to our understanding of the complex relationship between the Indus Civilization and climate change. New questions have also emerged about the relative importance of winter rain from the Indian Winter Monsoon (IWM) system and summer rain from the Indian Summer Monsoon (ISM) during the critical time period from 5.4 to 3.0 ka BP, which spans the pre-urban, urban, and post-urban phases of the Indus Civilization (Giosan et al., 2018; Petrie et al., 2017; Prasad and Enzel, 2006). This is because the winter rain zone partially overlaps with the summer rain zone (Figure 1), and provides a critical supply of rain and snowfall for the Indus River basin. However, we currently understand much less about the behavior of the IWM than the ISM.

At its height, the Indus Civilization spanned a considerable geographical area with a greater extent than the other ancient civilizations of its time (Agrawal, 2007; Possehl, 2003). Today, the region that was once occupied by Indus populations is marked by a heterogeneous rainfall pattern, and some locations in the central Thar desert receive as little as 100 mm yr$^{-1}$, which is only about 10% of the amount of direct annual rainfall compared to New Delhi. Scarce direct precipitation in the central regions around the Thar Desert is supplemented in some cases by fluvial or groundwater sources. In addition, the distribution of winter rain (increasing towards the northwest) is distinct from summer rain (increasing towards the east), making regions variably suitable for growing certain crops and grazing (Petrie et al., 2017; Petrie and Bates, 2017). While many paleoclimate studies from South Asia (references A-C, I, K-M, S, and U in Figure 1) have theorized about the overall climatic impact of drought (and in most cases identified summer monsoon as the cause), it is important to identify changes in the relative contributions and timing of seasonal rainfall from both the winter and summer monsoons. Previously, it has not been possible to reliably differentiate winter and summer rain in reconstructions from the Indus region.

In this study, we re-examined the same marine core (63KA) used in the original research of Staubwasser et al. (2003). We first assessed the reproducibility of the *Globigerinoides ruber* δ¹⁸O record using a larger size fraction of the same species for the time period 5.4-3.0 ka BP. We also measured the δ¹⁸O of two additional foraminifer species, *G. sacculifer* (*Globigerinoides sacculifer*) and *N. dutertrei* (*Neogloboquadrina dutertrei*), which live deeper than *G. ruber* in the water column. The different ecologies of the three species provide additional information with which to evaluate the multiple δ¹⁸O records and assess seasonal changes in the paleoceanography of the northeastern Arabian Sea near the mouth of the Indus River.

The δ¹⁸O of foraminifera has been widely applied as an indicator of temperature and salinity changes (Duplessy et al., 1992; Maslin et al., 1995; Wang et al., 1995; Rohling, 2000; among others). Measuring the δ¹⁸O of species calcifying at different depths can provide further information about upper ocean seasonal hydrography such as surface water mixing, depth of the thermocline, and upwelling (Ravelo and Shackleton, 1995). Similar methods have been applied by several other studies (Billups et al., 1999; Cannariato and Ravelo, 1997; Norris, 1998; Steinke et al., 2010; Steph et al., 2009; among others), including a reconstruction of East Asian Winter Monsoon strength in the South China Sea (Tian et al., 2005). Here we apply a comparable method to samples from core 63KA in the northeastern Arabian Sea because surface waters at this location are influenced by freshwater discharge from the Indus River and direct precipitation during the summer monsoon months, whereas enhanced upper ocean mixing occurs during the winter monsoon. We hypothesized that our new measurements of δ¹⁸O of *G. sacculifer* and *N. dutertrei* would allow us to track changes in upper ocean mixing. Weaker IWM winds are expected to result in a shorter duration and/or less intense upper ocean mixing, although how this signal is ultimately related to the amount or distribution of winter rainfall in the Indus River catchment has not been demonstrated conclusively. Dimri (2006) studied Western Disturbances for the time period 1958-1997, and noted that years of surplus winter precipitation are linked to significant heat loss over the northern Arabian Sea, which is mainly attributed to intensified westerly moisture flow and enhanced evaporation. Such conditions would promote deeper winter mixing, and provide a basis for relating thermocline depth with IWM intensity. By comparing the δ¹⁸O of multiple species of foraminifera we seek to infer variations in the relative strengths of the summer and winter monsoons, and by comparing the 63KA record to other nearby marine and terrestrial records we evaluate the potential role that climate played in cultural transformation of the Indus Civilization.

**2. Site Description**

*2.1 Monsoon – land-based processes*

Today, most of the annual precipitation over northwest South Asia stems from the ISM, and occurs mainly between June and September. The pressure gradient between the low-pressure Tibetan Plateau and high-pressure Indian Ocean is accompanied by the ITCZ (Intertropical Convergence Zone) reaching its northward maximum in summer, which draws in moisture over the subcontinent via southwesterly winds from the Indian Ocean (Gadgil, 2003). The summer rainfall gradient increases from the central Thar Desert (as little as 100

mm direct summer rainfall per year) to the Himalaya mountains in the north (>1000 mm) and
the Aravalli range to the west (>500 mm) (Figure 1b).
The IWM rain falls between December through March, and is mainly the result of atmospheric
Western Disturbances (Dimri and Dash, 2012; Yadav et al., 2012) originating over the
Mediterranean and Black Sea (Hatwar et al., 2005) that allow for moisture incursion from the
Arabian Sea (Rangachary and Bandyopadhyay, 1987). During the IWM, the pressure gradient
is reversed from the summer condition, allowing the passage of Western Disturbances when
the ITCZ moves southward. As winter transitions to spring, predominantly northeasterly
winds shift to westerly winds (Sirocko, 1991) that result in peak winter rainfall over the plains
of northwest India in February and March. Anomalously cool, evaporative conditions over the
northern Arabian Sea (promoting deeper winter mixing) also correlates with increased winter
precipitation in the western Himalayas (Dimri, 2006). The winter rainfall gradient increases
from the southern Thar Desert (<10 mm per year) up to the Himalayas in the northwest (>400
mm) (Figure 1c). Overall, the IWM contributes between roughly 10 to 50% of the total annual
rainfall of northwest South Asia today.

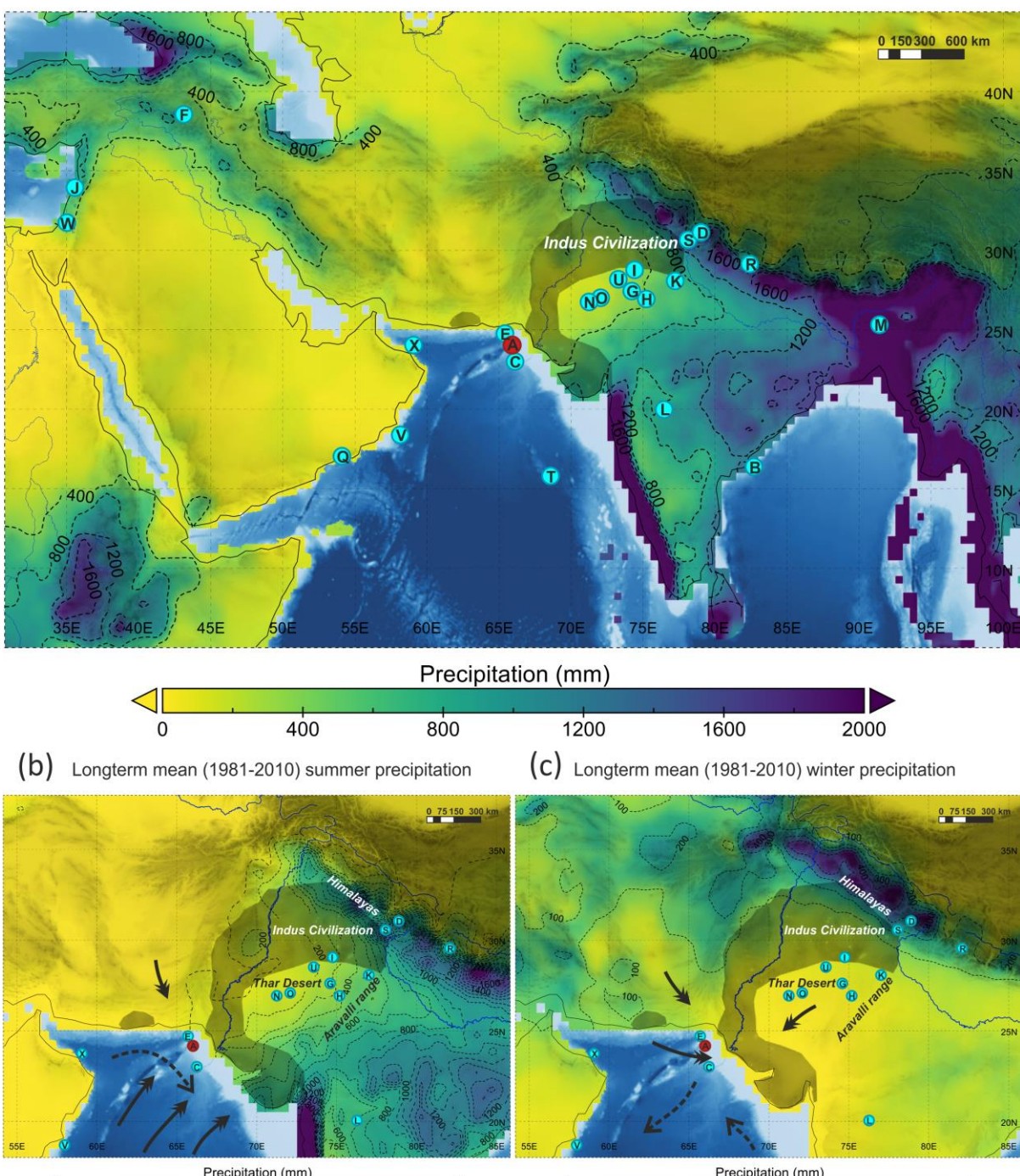

(a) Longterm mean (1981-2010) annual precipitation

Precipitation (mm)

(b) Longterm mean (1981-2010) summer precipitation

(c) Longterm mean (1981-2010) winter precipitation

Precipitation (mm)

Precipitation (mm)

**Figure 1. a.** Annual **b.** ISM (JJAS) **c.** IWM (DJFM) mean precipitation (1981-2010) isohyets taken from the GPCC V7 global gridded dataset (0.5° x 0.5° resolution) (Schneider et al., 2015); note the difference in scale for summer and winter precipitation (0-2000 mm vs. 0-500 mm). Rainfall data overlain on GEBCO 2014 ocean bathymetry dataset (Weatherall et al., 2015), and shaded region shows extent of the Indus Civilization. Bold arrows show main wind directions, dashed arrows show ocean surface currents. Other studies discussed in this paper indicated by letters:

| | | | |
|---|---|---|---|
| A | Core 63KA – (this study; Staubwasser et al., 2003) | E | Core 39KG and 56KA – (Doose-Rolinkski et al., 2001) |
| B | Core 16A – (Ponton et al., 2012) | F | Lake Van record – (Wick et al., 2003; Lemcke and |
| C | Core Indus 11C – (Giosan et al., 2018) | | Sturm, 1997) |
| D | Din Gad peat record – (Phadtare, 2000) | G | Didwana playa lake – (Singh et al., 1990) |

H   Sambhar playa lake – (Sinha et al., 2006)
I   Karsandi playa lake – (Dixit et al., 2018)
J   Jeita cave speleothem – (Cheng et al., 2015)
K   Kotla Dahar lake – (Dixit et al., 2014)
L   Lonar lake – (Menzel et al., 2014)
M   Mawmluh cave speleothem – (Berkelhammer et al.., 2012)
N   Kanod playa lake – (Deotare et al., 2004)
O   Bap Malar playa lake – (Deotare et al., 2004)
Q   Qunf cave speleothem – (Fleitmann et al., 2003)

R   Rara lake – (Nakamura et al., 2016)
S   Sahiya cave speleothem – (Kathayat et al., 2017)
T   Foraminifer trap EAST – (Curry et al., 1992)
U   Lunkaransar playa lake – (Enzel et al., 1999)
V   Core 723A, RC27-14, RC27-23, RC27-28 – (Gupta et al., 2003), (Overpeck et al., 1996)
W   Soreq cave speleothem – (Bar-Matthews et al., 2003; Bar-Matthews and Ayalon, 2011)
X   Core M5-422 – (Cullen et al., 2000)


The Indus and the other rivers that make up Punjab are partly fed by winter snow and ice melt
from their upper mountain catchment areas. Melting peaks during the summer months
around July-August (Yu et al., 2013), which coincides with the peak of ISM rainfall, and Indus
River discharge reaches its maximum during August (Karim and Veizer, 2002). The proportion
of winter to summer precipitation contributing to the Indus River is not entirely clear,
although one study has estimated a 64-72% contribution of winter precipitation from the
deuterium excess of Indus River water (Karim and Veizer, 2002), whereas a previous study
estimated a lower 15-44% contribution of snowmelt to Indus tributaries (Ramasastri, 1999).
Since the 1960s, the Indus River has seen more than a 50% reduction in discharge because of
the construction of barrages as well as the diversion of water for agricultural uses (Ahmad et
al., 2001).

*2.2 Hydrography – core site and ocean-based processes*


Core 63KA was obtained by the PAKOMIN cruise in 1993 (von Rad et al., 1995). The laminated
core from the northeastern Arabian Sea (24° 37' N, 65° 59' E) was taken at 316 m water depth
on the continental shelf, ~100 km west of the Indus River delta. The core has high
sedimentation rates (equivalent to a temporal resolution of around 18 years/cm in the period
of interest, 5.4-3.0 ka BP), and all foraminifer proxies were produced from the same laminated
core with no bioturbation. An important aspect of core 63KA is that different components of
the monsoon system are co-registered in the same sediment core, thereby permitting an
explicit evaluation of the relative timing of different parts of the climate system (e.g., ISM and
IWM).

Modern hydrographic conditions in the northeastern Arabian Sea are highly influenced by the
seasonal monsoon. During summertime, highest sea surface temperatures (SSTs) are
observed along with a shallow mixed layer depth <25 m (Schulz et al., 2002) (Figure 2a). A low
salinity plume surrounds the Indus River delta and shoreline extending as far as the coring
location (Supplemental Figure S1). The reverse occurs in winter when the lowest SSTs are
accompanied by surface water mixing to >125 m, resulting in warming of the deeper waters
(Schulz et al., 2002). Northeasterly winds promote convection in the northeastern Arabian
Sea by cooling and evaporation of surface water (Banse, 1984; Madhupratap et al., 1996), and
during the transition from winter to spring, wind directions shift from northeasterly to
westerly (Sirocko, 1991).

The northern Arabian Sea is dominated by highly saline (up to 37 psu) surface waters known
as Arabian Sea High Salinity Water (ASHSW), which extends from the surface down to 100 m
depth (Joseph and Freeland, 2005). The high salinity is explained by the high evaporative rates
over this region. ASHSW forms in the winter, but is prevented from reaching our coring site
on the shelf by northerly subsurface currents until the summer (Kumar and Prasad, 1999).
Along coastal areas, the ASHSW is starkly contrasted by the fresh water discharge of the Indus
River, combined with direct precipitation. In contrast, surface waters in the Bay of Bengal on
the eastern side of India have much lower surface water salinity, because of overall higher
precipitation and stronger stratification from weaker winds (Shenoi et al., 2002). The
heightened evaporative conditions and highly saline surface waters of the northeastern
Arabian Sea make it a sensitive study location to observe changes in discharge of the entire
Indus River catchment area – ultimately tracking changes in monsoon strength. Unlike
individual terrestrial records, which may be affected by local climatic processes, the marine
record from core 63KA is more likely to integrate regional changes of the large-scale ocean-
atmosphere system.
Planktonic foraminifera complete their life cycle within a few weeks (Bé and Hutson, 1977).
Peak abundances indicate the time of year when each species tends to calcify, thereby
recording the $\delta^{18}O$ and temperature of the seawater in their $CaCO_3$ shells primarily during
certain seasons. Foraminifer abundances in the eastern Arabian Sea have been studied by
Curry et al. (1992) using sediment traps deployed at shallow (~1400 m) and deep (~2800 m)
water depths ("T" in Figure 1a). *G. ruber* and *G. sacculifer* have peak abundances during the
summer months (June-September), whereas *N. dutertrei* lives mainly during the winter and
has a secondary peak in summer months (Figure 2c). Preferred depth ranges for each species
reflect their ecological niches, including requirements for nutrients and tolerance for ranges
of temperature and salinity (Bé and Hutson, 1977; Hemleben et al., 2012). *G. ruber* lives in
the upper surface waters (0-10 m), *G. sacculifer* is found in slightly deeper surface waters (10-
40 m), and *N. dutertrei* inhabits the base of the mixed layer near the thermocline (40-140 m)
(estimates based on ranges from Farmer et al. (2007) and the local CTD profiles) (Figure 2d).

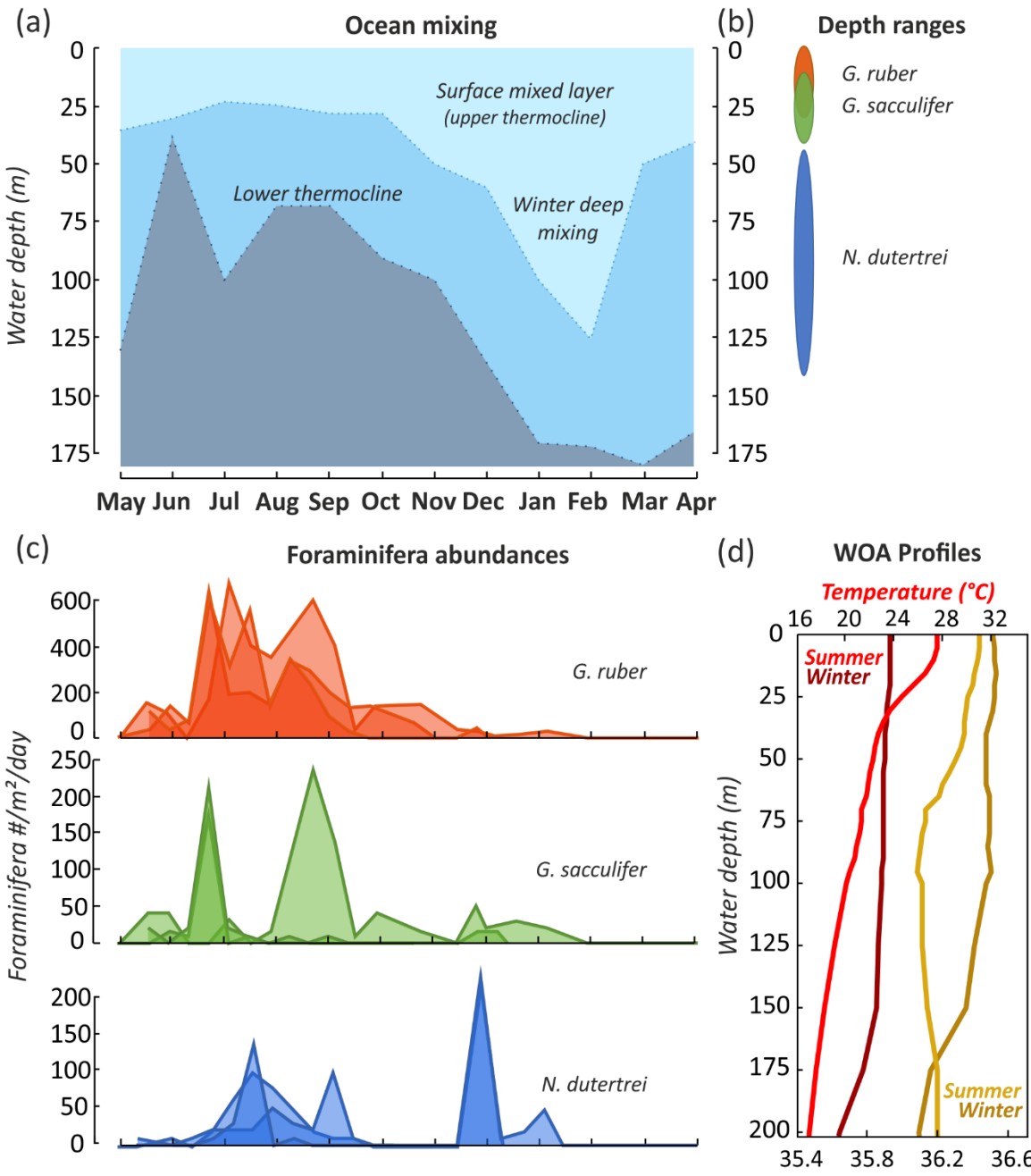

**Figure 2. a.** Seasonal surface water mixing depth based on station EPT-2 located nearby the coring site of 63KA (adapted from Schulz et al., 2002 who also used data from Hastenrath and Lamb, 1979) **b.** Foraminifer depth ranges based on CTD profile **c.** Foraminifer abundances from EAST traps (overlapping peaks indicate data from multiple traps): *G. ruber* (orange), *G. sacculifer* (green), and *N. dutertrei* (blue) (adapted from Curry et al., 1992 using Zaric, 2005) **d.** World Ocean Atlas (WOA) mean (1955-2012) temperature (red) and salinity (yellow) profiles at 24.875°N, 65.875°E, shown for summer (JAS) and winter (JFM) seasons (Locarnini et al., 2013; Zweng et al., 2013).

## 3. Materials and Methods

*3.1 Age model*

The radiocarbon dates from Staubwasser et al. (2002, 2003) were obtained from 80 samples
of mainly the foraminifer *G. sacculifer* and three samples of *O. universa*. In the interval of
interest (5.4-3.0 ka BP), there are 15 radiocarbon dates with a 95% confidence range of 30-
130 years. The average sample resolution is 18 years/cm. Bayesian age modelling software,
BACON v2.3.3 (Blaauw and Christen, 2011), was used as an R-package to update the age
model of core 63KA. No major difference exists between the old and new age models, except
for the period 13-11 ka BP (Supplemental Figure S5, Table S2). IntCal13 was used for
radiocarbon calibration (Reimer et al., 2013) with marine reservoir ages provided by
Staubwasser et al. (2002, 2003).
*3.2 Stable isotope analysis*
Oxygen and carbon isotopes were measured on three species of foraminifera selected from
washed samples at 1-cm intervals throughout 132 cm of the core covering 5.4-3.0 ka BP: *G.*
*ruber* (white, *sensu stricto*), *G. sacculifer*, and *N. dutertrei*. For *G. ruber*, 12 ± 8 foraminifera
were picked from the 400-500μm size fraction with an average weight of 21.4 ± 2.5μg. The
400-500μm size fraction was picked because too few specimens remained in the size fraction
315-400μm used by Staubwasser et al. (2003). For *G. sacculifer*, 34 ± 7 foraminifera were
picked from the 315-400μm size fraction with an average weight of 21.9 ± 2.6μg. For *N.*
*dutertrei*, 34 ± 4 foraminifera were picked from the 315-400μm size fraction with an average
weight of 25.9 ± 2.2μg. At some depth levels in the core there were insufficient foraminifera
for measurement, along with outlier measurements in two cases, leaving 14 gaps in the *G.*
*ruber* 400-500μm record, 4 gaps in the *G. sacculifer* record, and no gaps for *N. dutertrei*. The
published *G. ruber* is from the 315-400μm size fraction and contains 17 gaps in the depth
range examined (Staubwasser et al., 2003).
All foraminifera were weighed, crushed, and dried at 50° C. Samples were cleaned for 30
minutes with 3% $H_2O_2$, followed by a few drops of acetone, ultrasonication, and drying
overnight. Where sample weights exceeded 80μg, oxygen and carbon isotopes were
measured using a Micromass Multicarb Sample Preparation System attached to a VG SIRA
Mass Spectrometer. In cases of smaller sample sizes, the Thermo Scientific Kiel device
attached to a Thermo Scientific MAT253 Mass Spectrometer was used in dual inlet mode. This
method adds 100% $H_3PO_4$ to the $CaCO_3$, water is removed cryogenically, and the dry $CO_2$ is
analyzed isotopically by comparison with a laboratory reference gas. For both measurement
methods, 10 reference carbonates and 2 control samples were included with every 30
samples. Results are reported relative to VPDB, and long-term reproducibility of laboratory
standards (e.g., Carrara marble) is better than ±0.08‰ for $\delta^{18}O$ and ±0.06‰ for $\delta^{13}C$.
Reproducibility of foraminiferal measurements was estimated by five triplicate (three
separately picked) measurements of *G. ruber* (400-500μm) that yielded one standard
deviation of ±0.12‰ ($\delta^{18}O$) and ±0.10‰ ($\delta^{13}C$). For *G. sacculifer* (315-400μm) the standard
deviation of eight triplicate measurements was ±0.07‰ ($\delta^{18}O$) and ±0.07‰ ($\delta^{13}C$), and for *N.*
*dutertrei* (315-400μm) the standard deviation of nine triplicate measurements was ±0.06‰
($\delta^{18}O$) and ±0.07‰ ($\delta^{13}C$).
To calculate equilibrium values of $\delta^{18}O_{calcite(PDB)}$, we used the CTD profile from station 11
(24.62° N, 66.07° E) taken in September 1993 during PAKOMIN *Sonne* cruise no. 90 (von Rad,
2013), which is nearly identical to the location of core 63KA (24.62° N 65.98° E). The
$\delta^{18}O_{water(SMOW)}$ was calculated from salinity following Dahl and Oppo (2006), and
$\delta^{18}O_{calcite(SMOW)}$ was further calculated using the calcite-water equation of Kim and O'Neil
(1997). We also used the equation of Shackleton (1974) as a comparative method for
calculating $\delta^{18}O_{calcite(PDB)}$.
*3.3 Statistical treatment*
Statistical tests were applied to the raw data from the $\delta^{18}O$ and $\delta^{13}C$ time series, including the
package SiZer (Chaudhuri and Marron, 1999; Sonderegger et al., 2009) in R software (2016)
that calculates whether the derivative of a time series exhibits significant changes given a
range of timespans. A Pearson's correlation test (confidence level 95%) was done on paired
samples from both size fractions of *G. ruber*. We also conducted a Welch's t-test to determine
if the mean population of $\delta^{18}O$ is significantly different before and after 4.1 ka BP.
As in the original data of Staubwasser et al. (2003), the oxygen isotope results show great
variability and distinguishing long-term trends in these data benefits from smoothing for
visualization purposes. After completing all statistical tests and performing the differences on
the raw data (132 depths), a loess (locally weighted) smoothing function was applied to the
$\delta^{18}O$ and $\delta^{13}C$ data from 5.4-3.0 ka BP, using a 210-year moving window as described by
Staubwasser et al. (2003). Loess smoothing uses weighted least squares, which places more
importance on the data points closest to the center of the smoothing interval. The bandwidth
of 210 years was considered a reasonable time window for capturing the overall trends in the
dataset (other time windows are shown for comparison in Supplemental Figure S2).
**4. Results**
The new $\delta^{18}O$ measurements of *G. ruber* (400-500μm) parallel the published record of *G.*
*ruber* (315-400μm) (Staubwasser et al., 2003), but the $\delta^{18}O$ of the specimens from the larger
size fraction is offset by -0.23‰ on average (Figure 3). The records from two size fractions,
produced in different laboratories by different investigators, display a weak positive
correlation for the raw data (R = 0.25, p < 0.01, n = 109, slope 0.26, intercept -1.36), and the
210-year smoothed records reveal good agreement in the overall trends of the data. When
comparing the two *G. ruber* records, it is apparent that the increasing trend in $\delta^{18}O$ starts well
before ~4.2 ka BP – perhaps as early as ~4.9 ka BP. This trend is also observed with the SiZer
analysis, which identifies a significant increase in $\delta^{18}O$ anywhere from 4.9 to 4.2 ka BP
depending on which smoothing window is selected (Figure 4). The new $\delta^{18}O$ record of *G. ruber*
*(*400-500μm) shows additional detail after the ~4.2 ka BP event – i.e. specifically, a double-
peak maximum occurring at 4.1 and 3.95 ka BP that is related to seven discrete measurements
with high $\delta^{18}O$ values. These maxima are offset from the average $\delta^{18}O$ value by +0.18‰
(smoothed average), or up to +0.38‰ when considering the maximum individual
measurement at 4.1 ka BP. The offsets from the average values exceed one standard
deviation of the entire record from 5.4-3.0 ka BP, which is 0.13‰. Although *G. ruber* shows
an event at 4.1 ka BP, it does not show a permanent step change: A Welch's t-test comparing
the means of pre- and post-4.1 ka BP indicates that the +0.07‰ shift in mean $\delta^{18}O$ values of
*G. ruber* (315-400μm) is statistically significant (t value = 2.9, p < 0.01, n = 115), but the
+0.03‰ shift in mean $\delta^{18}O$ values of *G. ruber* (400-500μm) is not significant (t value = 1.5, p
< 0.2, n = 118).

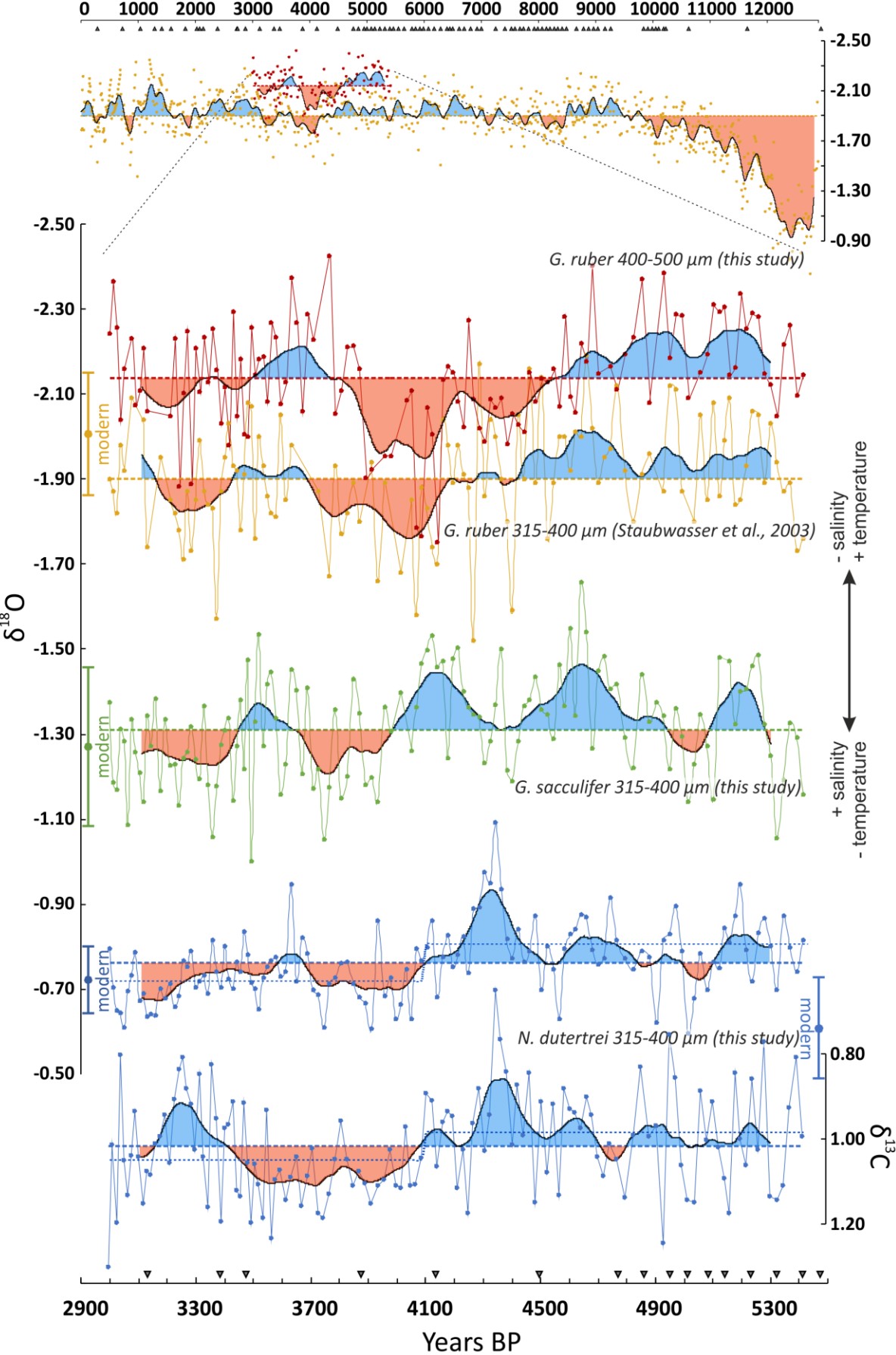

**Figure 3.** Core 63KA δ¹⁸O *G. ruber* from two size fractions: 400-500μm (red) (this study), 315-400μm (orange) (Staubwasser et al., 2003), shown in the context of the original record and also zoomed in over 5.4-3.0 ka BP. δ¹⁸O of *G. sacculifer* 315-400μm (green), and δ¹⁸O and δ¹³C of *N. dutertrei* 315-400μm (blue) are shown over the interval 5.4-3.0 ka BP. Data are shown with a 210-year loess smoothing, and modern surface values ±1σ are plotted for comparison. Mean values for all species are denoted by the dotted line, and the pre- and post-4.1 ka BP mean values are indicated by an additional dotted line for *N. dutertrei*. Individual AMS radiocarbon dates are denoted by triangles near the timeline.

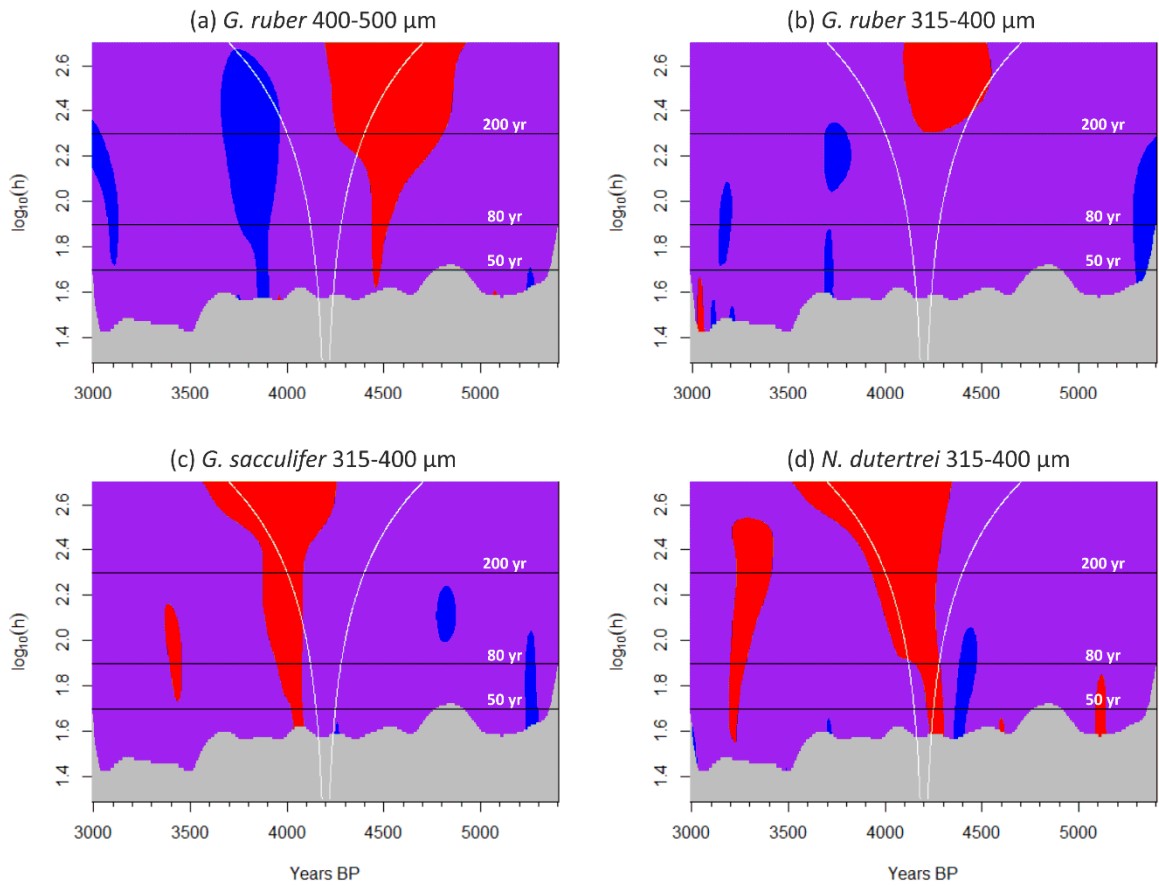

**Figure 4.** SiZer 1ˢᵗ derivative analysis (Chaudhuri and Marron, 1999; Sonderegger et al., 2009) applied to δ¹⁸O of **a.** *G. ruber* 400-500μm, **b.** *G. ruber* 315-400μm, **c.** *G. sacculifer* 315-400μm, **d.** *N. dutertrei* 315-400μm. The red areas indicate statistically significant increases in δ¹⁸O, the blue represent decreases, and the purple no significant change. Black horizontal lines are the smoothing bandwidths (h = 50, 80, and 200 years). The distance between the white lines denotes the change in smoothing bandwidth scaled to the x-axis.

The relative differences in δ¹⁸O of the planktonic species studied (*G. ruber*, *G. sacculifer* and *N. dutertrei*) reflect the temperature and salinity of their habitat in the water column: δ¹⁸O *G. ruber* < δ¹⁸O *G. sacculifer* < δ¹⁸O *N. dutertrei* (Figure 3). *G. sacculifer* is offset from *G. ruber* (315-400μm) by approximately +0.57‰, whereas *N. dutertrei* is offset by +1.14‰. The larger size fraction of *G. ruber* (400-500μm) is offset from *G. ruber* (315-400μm) by -0.23‰. The offsets among species are maintained throughout the entire record (Figure 3). We also measured δ¹⁸O values near the top of the core (approximately the last 200 years) for all three species in the 315-400μm size fraction, which continue to show the same offsets

(Supplemental Figure S3). The $\delta^{18}$O of *G. ruber* shows the greatest variance and *N. dutertrei*
shows the least (Supplemental Figure S4, Table S1).
Equilibrium calcite calculations based on the salinity and temperature measurements from
the September 1993 CTD profile of station 11 of the PAKOMIN Cruise (von Rad, 2013) show
the expected depth habitats of the three foraminifer species (Figure 5). *G. ruber* is generally
found at 0-30 m, *G. sacculifer* at 15-40 m, and *N. dutertrei* at 60-150 m (Farmer et al., 2007).
Using the CTD profile from our core location, we compare these depth ranges with the
measured $\delta^{18}$O values. The calculated depths ranges agree well with those expected on the
basis of other studies, placing *G. ruber* in the upper 10 m, *G. sacculifer* 10-40 m, and *N.*
*dutertrei* 40-140 m.

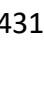

**Figure 5.** $\delta^{18}$O of equilibrium calcite (left) calculated from the CTD temperature and salinity profile at
station 11 (von Rad, 2013) (right) with projected depth ranges of *G. ruber* 400-500μm (red), *G. ruber*
315-400μm (orange), *G. sacculifer* 315-400μm (green), *N. dutertrei* 315-400μm (blue). We show
estimated values using both the original paleotemperature equation of Shackleton (1974) (dark teal),
and Kim & O'Neil (1997) (turquoise). Horizontal ranges show the measured $\delta^{18}$O values of each species
between 5.4-3.0 ka BP.

*G. sacculifer* $\delta^{18}$O increases around 4.1 ka BP, and a Welch's t-test comparing the means of
pre- and post-4.1 ka BP indicates that the +0.08‰ shift in mean $\delta^{18}$O values is statistically
significant (t value = 3.8, p < 0.01, n = 128). SiZer analysis also points to a statistically significant
increase at ~4.1-3.9 ka BP, when considering all smoothing time windows between 20 and
500 years (Figure 4).

Likewise, the dominant change in the $\delta^{18}$O of *N. dutertrei* is a mean increase at 4.1 ka BP
(Figure 3). SiZer analysis also identifies a significant decrease in $\delta^{18}$O occurring mainly
between 4.45 and 4.35 ka BP, followed by a significant increase between 4.3 and 4.1 ka BP
(Figure 4). A Welch's t-test comparing the means of pre- and post-4.1 ka BP indicates that the
+0.08‰ shift in mean $\delta^{18}$O values is statistically significant (t value = 6.2, p < 0.01, n = 132),
along with the +0.07‰ shift in mean $\delta^{13}$C (t value = 3.3, p < 0.01, n = 132).

Differencing $\delta^{18}$O of foraminifera (expressed as $\Delta\delta^{18}$O) in the same sample can better
emphasize signals of interest (Figure 6). The $\Delta\delta^{18}$O of *G. ruber* 400-500μm and *G. ruber* 315-
400μm size fractions shows increasing similarity between ~4.8 and 3.9 ka BP during the period
of overall higher $\delta^{18}$O. The $\Delta\delta^{18}$O of *N. dutertrei* and both size fractions of *G. ruber*, designated
$\Delta\delta^{18}$O$_{d-r}$, reveals a period of more similar values between ~4.5 and 3.9 ka BP, with two minima
at 4.3 and 4.1 ka BP. The $\Delta\delta^{18}$O of *G. sacculifer* and both size fractions of *G. ruber* ($\Delta\delta^{18}$O$_{s-r}$)
show a period of similar values between 4.3 and 3.9 ka BP, with a minimum difference at 4.1
ka BP. In contrast, the $\Delta\delta^{18}$O of *N. dutertrei* and *G. sacculifer* ($\Delta\delta^{18}$O$_{d-s}$) shows the most
similarity between 4.5 and 4.2 ka BP with a minimum at 4.3 ka BP, followed by the maximum
differences between 4.2 and 3.9 ka BP that peaks at 4.1 ka BP.

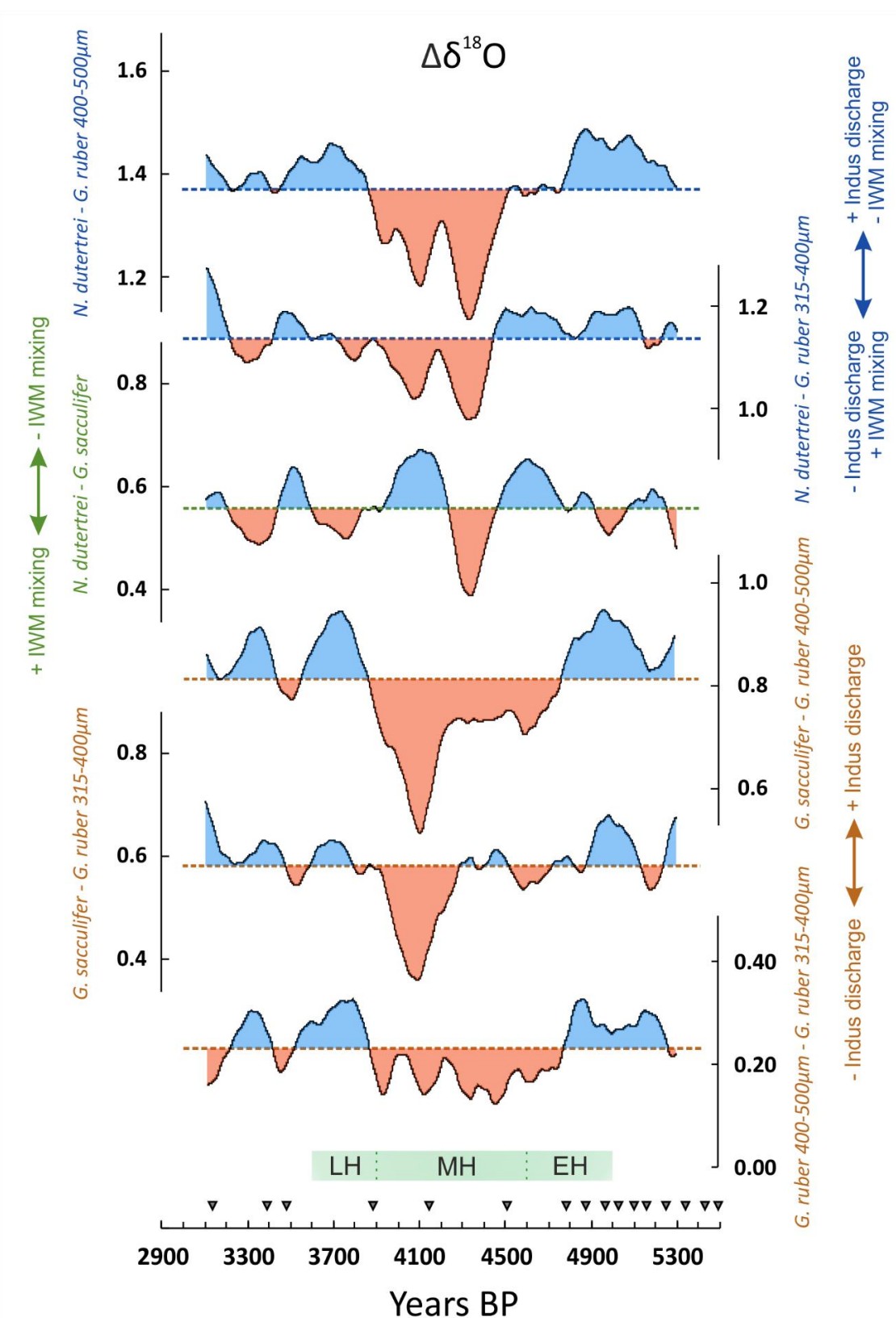

**Figure 6.** Core 63KA Δδ$^{18}$O shown with a 210-year loess smoothing. Individual AMS radiocarbon dates are denoted by triangles near the timeline. *G. ruber* 315-400μm size fraction data come from Staubwasser et al. (2003). The green band near the timeline showing EH, MH, and LH refers to Early Harappan (~5.0-4.6 ka BP), Mature Harappan (~4.6-3.9 ka BP), and Late Harappan (~3.9-3.6 ka BP) periods, respectively.

**5. Discussion**

*5.1 Interpretation of foraminifer δ$^{18}$O*

The trends in the original δ$^{18}$O record of *G. ruber* (315-400μm) by Staubwasser et al. (2003) is reflected by our independent δ$^{18}$O measurements of *G. ruber* in a larger size fraction (400-500μm), although an important difference exists suggesting a decrease in freshwater discharge as early as 4.8 ka BP. The larger size fraction is offset by approximately -0.2‰, which is similar to the size-related fractionation of -0.3‰ per +100μm for *G. ruber* reported by Cayre and Bassinot (1998), and could be attributed to size-related vital effects. Alternatively, part of the offset might be explained by interlaboratory calibration considering the data were produced using two different methods and mass spectrometers.

The observed 4.1 ka BP maximum in δ$^{18}$O of *G. ruber*, living near the surface during summer months, could be attributed to either decreased SST or increased surface water salinity (Bemis et al., 1998). Staubwasser et al. (2003) acknowledged that a decrease in SST could cause the increase in δ$^{18}$O in the *G. ruber* record, but argued that this explanation is unlikely because a *G. ruber* δ$^{18}$O record from core M5-422 in the northwestern Arabian Sea shows opposing trends over the same time period (Cullen et al., 2000), and a local alkenone SST proxy record shows relatively higher temperatures in the same period (Doose-Rolinski et al., 2001). If the ~0.2‰ (relative to mean) increase in δ$^{18}$O of *G. ruber* at 4.1 ka BP was caused by temperature change rather than salinity, a ~1° C cooling of surface water would be required (Kim and O'Neil, 1997).

Following Staubwasser et al. (2003), we interpret the δ$^{18}$O variations of *G. ruber* to be predominantly a salinity signal. Salinity at the core site is dependent on changes in Indus River discharge, local run-off, and direct precipitation. Although the ISM would be the main influence on direct precipitation and run-off at the coring location, changes in the IWM could also influence Indus River discharge.

The thermocline-dwelling foraminifera *N. dutertrei* shows maximum abundances during winter, and are interpreted to reflect winter mixing. During weak IWM conditions, colder unmixed water would result in higher δ$^{18}$O values of *N. dutertrei*, whereas enhanced mixing and homogenization of the water column under strong IWM conditions would decrease δ$^{18}$O. The minimum of δ$^{18}$O in *N. dutertrei* occurs between 4.5 and 4.3 ka BP, pointing to a period of strengthened IWM. We interpret the stepped increase in δ$^{18}$O of *N. dutertrei* at 4.1 ka BP to represent a decrease in IWM wind-driven mixing. Similarly, δ$^{13}$C of *N. dutertrei* increases significantly after 4.1 ka BP (Figure 3), which could indicate reduced upwelling of low δ$^{13}$C intermediate water (Lynch-Stieglitz, 2006; Ravelo and Hillaire-Marcel, 2007; Sautter and Thunell, 1991); however, the interpretation of δ$^{13}$C remains uncertain because of a poor understanding of the controls on the δ$^{13}$C of planktonic foraminifera in this region. According to the δ$^{18}$O signal of *N. dutertrei*, the temperature pattern in the thermocline implies surface cooling between 4.5 and 4.3 ka BP and surface warming after 4.1 ka BP interrupted only by a period of cooling between 3.7 and 3.3 ka BP, which is in broad agreement with records of alkenone sea-surface temperature estimates from cores in the northeastern Arabian Sea ("E" in Figure 1) (Doose-Rolinski et al., 2001; Staubwasser, 2012).

          *5.2 Interpretation of foraminifer $\Delta\delta^{18}O$*


By using $\Delta\delta^{18}O$ between foraminifer species, we can distinguish additional processes affecting
the surface waters and thermocline (Ravelo and Shackleton, 1995). This technique has been
used previously to infer changes in the strength of the East Asian Winter Monsoon (EAWM)
in the South China Sea (Tian et al., 2005), as well as mixed layer and thermocline depth in
other studies (Billups et al., 1999; Cannariato and Ravelo, 1997; Norris, 1998). Here we use
the difference in the $\delta^{18}O$ of *G. ruber* and *N. dutertrei* ($\Delta\delta^{18}O_{d-r}$) to track changes in the
surface-to-deep gradient. This gradient can be driven by either $\delta^{18}O$ changes in the surface-
dwelling (*G. ruber*) and/or the thermocline-dwelling species (*N. dutertrei*). During times of a
strengthened winter monsoon, $\Delta\delta^{18}O_{d-r}$ will decrease as surface waters are homogenized and
the thermocline deepens. Similarly, $\Delta\delta^{18}O_{d-r}$ will also decrease during times of a weakened
summer monsoon, as decreased Indus River discharge will increase surface water salinity and
$\delta^{18}O$ of *G. ruber* will become more similar to *N. dutertrei*.

*G. sacculifer* is also a surface dweller, but has a slightly deeper depth habitat than *G. ruber*.
We thus expect *G. ruber* to be more influenced by surface salinity variations than *G. sacculifer*,
and suggest the $\delta^{18}O$ difference between the two species ($\Delta\delta^{18}O_{s-r}$) reflects the influence of
Indus River discharge on near surface salinity. The smallest difference in $\Delta\delta^{18}O_{s-r}$ occurs at 4.1
ka BP, which is interpreted as an increase in surface water salinity (Figure 6).

The difference in $\delta^{18}O$ between *G. sacculifer* and *N. dutertrei* ($\Delta\delta^{18}O_{d-s}$) also reflects surface
mixing and thermocline depth, but *G. sacculifer* is less affected by surface salinity changes
than *G. ruber*. Thus, the responses of $\Delta\delta^{18}O_{s-r}$ and $\Delta\delta^{18}O_{d-s}$ can be used to differentiate
between surface water salinity changes and wind-driven mixing. Accordingly, simultaneously
low $\Delta\delta^{18}O_{d-s}$ and $\Delta\delta^{18}O_{d-r}$ indicate a period of increased surface water mixing and increased
IWM (such as the period between 4.5 and 4.3 ka BP), but times of relatively low $\Delta\delta^{18}O_{d-s}$ but
high $\Delta\delta^{18}O_{d-r}$ and $\Delta\delta^{18}O_{s-r}$ (around 5.0 ka BP) indicate periods of increased Indus discharge and
strength of the ISM and IWM.

The following period of low $\Delta\delta^{18}O_{d-r}$ from 4.1-3.9 ka BP is likely driven by increased salinity of
surface water. This distinction becomes clearer when examining the $\Delta\delta^{18}O_{s-r}$, where increased
similarity from 4.8-3.9 ka BP (with a sharp increase at 4.1 ka BP) reflects the effect of increased
sea surface salinity that reduces the $\delta^{18}O$ difference between *G. ruber* and *G. sacculifer*. At
the same time, weakened winter mixing increases $\Delta\delta^{18}O_{d-s}$, which occurs from 4.2-3.9 ka BP.
Importantly, the proxies also indicate that increased IWM mixing is generally positively
correlated with increased Indus discharge, and vice versa. The single time period when this
does not hold true is 4.5-4.25 ka BP, when increased IWM mixing is coupled with decreased
Indus discharge.

In summary, our multi-species approach using $\delta^{18}O$ of *G. ruber*, *G. sacculifer*, and *N. dutertrei*
allows us to differentiate between strength of the IWM and freshwater discharge of the Indus
River. We suggest that ISM strength decreased gradually from at least 4.8 ka BP, while the
IWM strength peaked around 4.5-4.3 ka BP and then weakened afterwards. It is unlikely that
the abrupt increase in *G. ruber* $\delta^{18}O$ at 4.1 ka BP and low $\Delta\delta^{18}O_{s-r}$ could be caused solely by
the decrease in IWM strength, even though IWM contributes to Indus River discharge.
Weakening of the ISM must have played a substantial role in the 4.1 ka BP shift as well,
indicated by the period 4.5-4.25 ka BP, when Indus discharge reflected a weak ISM ($\Delta\delta^{18}O_{s-r}$)
despite a phase of strengthened IWM.

*5.3 Comparison to marine records*


Other marine records from the Arabian Sea also suggest a gradual decrease in ISM strength
from ~5 ka BP (Gupta et al., 2003; Overpeck et al., 1996). Cullen et al. (2000) observed an
abrupt peak in aeolian dolomite and calcite in marine sediments in the Gulf of Oman from
4.0-3.6 ka BP, and Ponton et al. (2012) also showed a shift to weaker ISM after 4.0 ka BP in
the Bay of Bengal, based on $\delta^{13}C$ of leaf waxes. Marine IWM reconstructions are not
particularly coherent: although Doose-Rolinski et al. (2001) find a decrease in evaporation
and weakening of the ISM between 4.6 and 3.7 ka BP, they argue this was accompanied by a
relative increase in IWM strength. Giosan et al. (2018) inferred enhanced winter monsoon
conditions from 4.5-3.0 ka BP based on a planktic paleo-DNA and % *Globigerina falconensis*
record close to our coring site ("C" in Figure 1), which disagrees with our finding of decreased
upper ocean mixing after 4.3 ka BP. We suggest that the high stratigraphic (i.e., laminated)
and chronological (i.e., 15 radiocarbon dates between 5.4-3.0 ka BP) resolution of core 63KA
paired with a multi-species foraminifer $\delta^{18}O$ record provides a robust history of the timing of
changes in IWM and ISM strength, but additional studies are needed to resolve some of the
discrepancies among the records.

*5.4 Comparison to regional terrestrial records*


The 63KA $\delta^{18}O$ record obtained from three foraminifer species highlights several important
ocean-atmosphere changes over the 5.4-3.0 ka BP time period. First, a sharp decrease
occurred in both summer and winter precipitation at 4.1 ka BP, which is within a broader 300-
year period of increased aridity spanning both rainfall seasons between 4.2 and 3.9 ka BP. In
detail, we infer a relative decrease in Indus River discharge and weakened ISM between 4.8
and 3.9 ka BP, peaking at 4.1 ka BP, while a 200-year-long interval of strong IWM interrupted
this period from 4.5-4.3 ka BP. Furthermore, the stepped change in $\delta^{18}O$ of *N. dutertrei*
suggests an enduring change in ocean-atmosphere conditions after 4.1 ka BP.

A relatively abrupt ~4.2 ka BP climate event has been observed in several terrestrial records
on the Indian subcontinent, most notably Mawmluh Cave (~4.1-3.9 ka BP) in northeastern
India (Berkelhammer et al., 2012) and Kotla Dahar (~4.1 ka BP) in northwestern India (Dixit et
al., 2014) (Figure 7). A less abrupt yet still arid period is documented in a peat profile (~4.0-
3.5 ka BP) from northcentral India (Phadtare, 2000), at Lonar Lake (~4.6-3.9 ka BP) in central
India (Menzel et al., 2014), and at Rara Lake (~4.2-3.7 ka BP) in western Nepal (Nakamura et
al., 2016). Finally, a recent study of oxygen and hydrogen isotopes in gypsum hydration water
from Karsandi on the northern margin of the Thar Desert showed wet conditions between 5.1
and 4.4 ka BP, after which the playa lake dried out sometime between 4.4 and 3.2 ka BP (Dixit
et al., 2018). Considering terrestrial records can record more local climatic conditions than
marine records, it is remarkable that the records collectively agree on a period of regional
aridity between 4.2 and 3.9 ka BP within the uncertainties of the age models that vary
considerably among records.

However, not all records support this finding. For example, a reconstruction from Sahiya Cave
in northwestern India shows an abrupt decrease in $\delta^{18}O$ interpreted to reflect an increase in
monsoon strength from ~4.3-4.15 ka BP, followed by an arid trend after 4.15 ka BP (Kathayat
et al., 2017). In addition, several other Thar Desert records do not identify a "4.2 ka BP event"
*sensu stricto*, but instead suggest that lakes dried out several centuries earlier (Deotare et al.,
2004; Enzel et al., 1999; Singh et al., 1990) or later (Sinha et al., 2006) than 4.2 ka BP. This
discrepancy may relate to non-linear climate responses of lakes, which would not record a
drought at 4.2 ka BP if they had already dried out earlier from the ongoing decrease in
summer rainfall. In addition, there are also significant concerns about chronological
uncertainties from the use of radiocarbon of bulk sediment for dating in some of these
records. It is also possible that variations in the timing of climate change inferred from the
terrestrial records may be real, reflecting different sensitivity to ISM and IWM rain. As a
marine record, core 63KA integrates large-scale ocean-atmosphere changes, and therefore
can help inform the interpretation of the more locally sensitive terrestrial records.
More distantly, several terrestrial records in the Middle East also show a decrease in winter
precipitation proxies around 4.2 ka BP: Jeita Cave in Lebanon records a relatively dry period
between 4.4 and 3.9 ka BP (Cheng et al., 2015) and Soreq Cave in Israel shows a period of
increased aridity starting at ~4.3 ka BP (Bar-Matthews et al., 2003; Bar-Matthews and Ayalon,
2011) (Figure 8). Lake Van in eastern Turkey also records reduced spring rainfall and enhanced
aridity after ~4.0 ka BP (Wick et al., 2003; Lemcke and Sturm, 1997). All of these records
suggest a relatively arid period with reduced winter precipitation after ~4.3 ka BP, as inferred
from core 63KA. Qunf Cave in Oman (Fleitmann et al., 2003), which is outside the range of
IWM influence, instead shows a steady mid-Holocene weakening of the ISM that closely
follows trends in summer solar insolation.

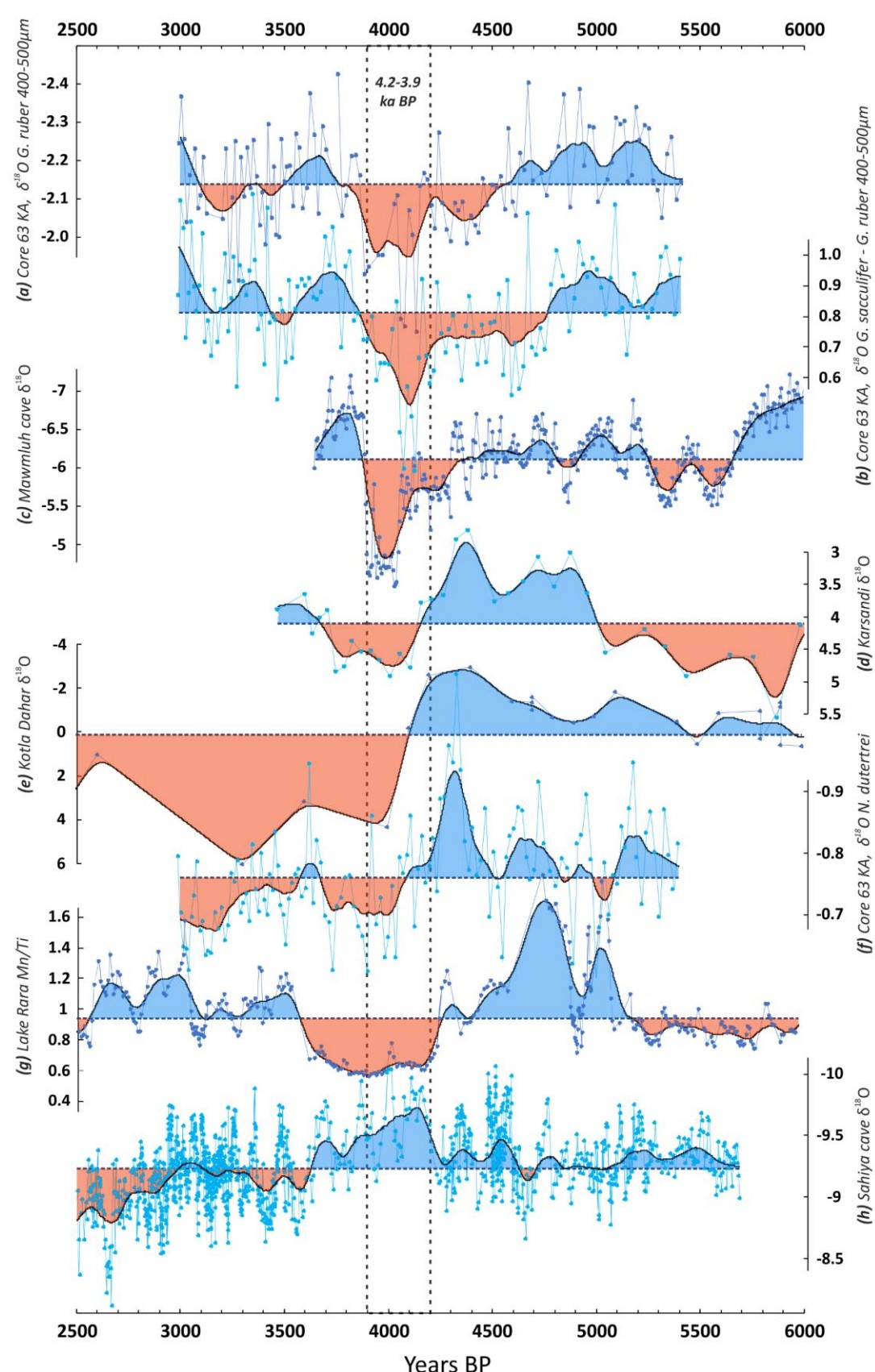

**Figure 7.** Comparison of the δ18O record of core 63KA with terrestrial records from the Indian
Subcontinent, from top to bottom: **a.** and **b.** this study; **c.** Berkelhammer et al., 2012; **d.** Dixit et al.,
2018; **e.** Dixit et al., 2014; **f.** this study; **g.** Nakamura et al., 2016; **h.** Kathayat et al., 2017. The mean

value for each record indicated by the horizontal dashed lines is taken for all available data between 6.0-2.5 ka BP.

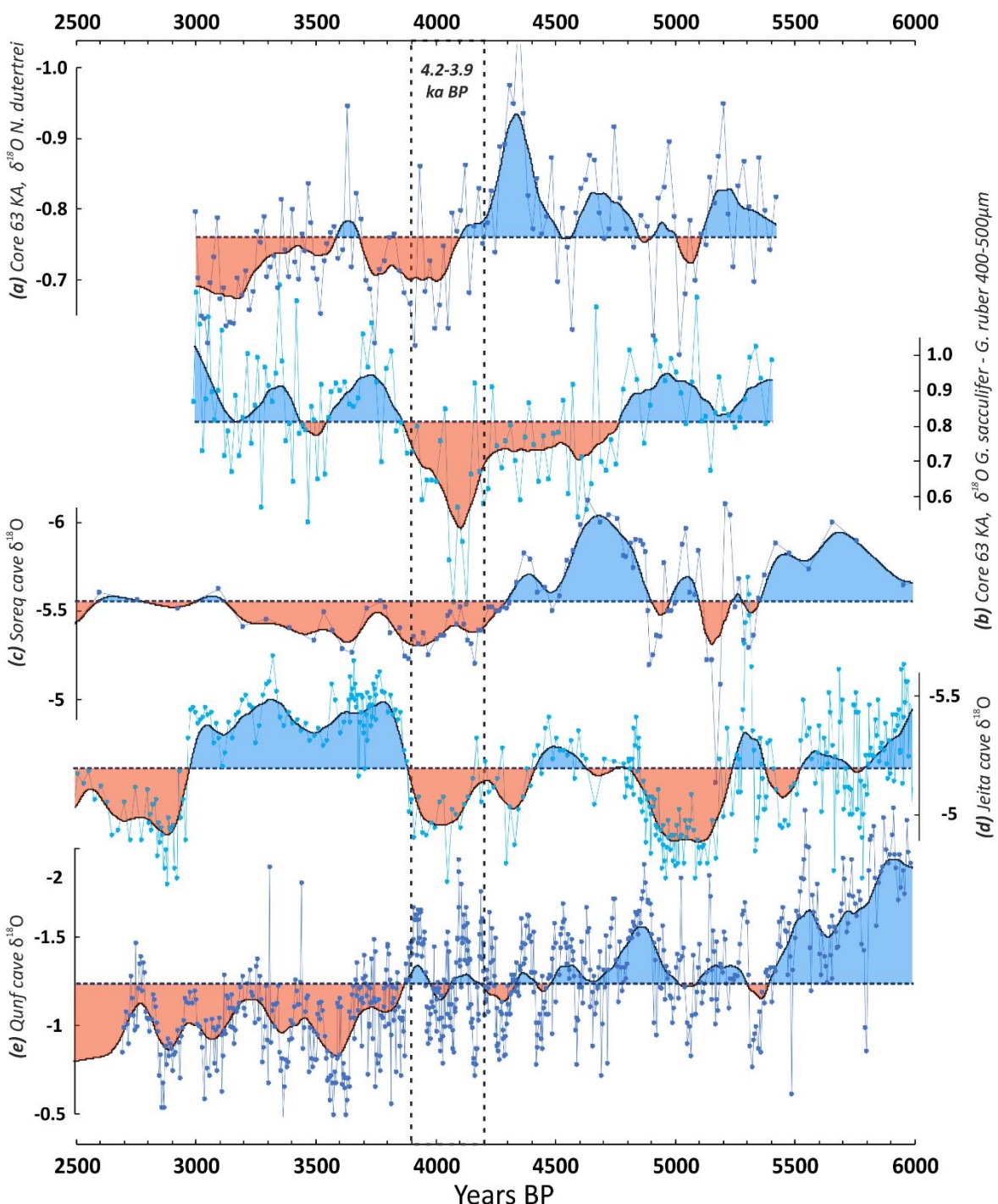

**Figure 8.** Comparison of the δ¹⁸O record of core 63KA with more distant records: **a.** and **b.** this study; **c.** Bar-Matthews et al., 2003; **d.** Cheng et al., 2015; **e.** Fleitmann et al., 2003. The mean value for each record indicated by the horizontal dashed lines is taken for all available data between 6.0-2.5 ka BP.

*5.5 Cultural impacts*

On the basis of our reconstruction of reduced IWM mixing after 4.3 ka BP, accompanied by decreased freshwater discharge of the Indus River, it is worth considering what impacts could be expected from a reduction in IWM and ISM precipitation. A weakened IWM overlying a reduced or more variable ISM would likely result in a distinct climate signal over the Indus River catchment, with broad implications for seasonal river flow and water availability throughout the year. The presence of the two rainfall systems creates a complex and diverse range of environments and ecologies across northwest South Asia (Petrie et al., 2017). In a situation when rainfall in both seasons is reduced over extended periods, step-shifts in the natural environment may occur that are difficult to reverse (e.g., desertification, lake desiccation, regional vegetation changes, decline in overbank flooding and shift in river avulsion patterns).

Societies reliant on IWM, ISM, or a combination of the two would have been vulnerable to years with monsoon failure, and a shift affecting both seasons will have challenged resilience and tested sustainability (Green and Bates et al. in press; Petrie et al., 2017). Archaeological research into the transition from the urban Mature Harappan phase (~4.6-3.9 ka BP) to the post-urban Late Harappan phase (~3.9-3.6 ka BP) notes progressive deurbanization through the abandonment of large Indus cities and a depopulation of the most western Indus regions, concurrent with a general trend towards an increase of concentrations of rural settlements in some areas of the eastern Indus extent (Green and Petrie, 2018; Petrie et al., 2017; Possehl, 1997) (Figure S6). The relatively limited range of well-resolved available archaeobotanical data suggests that there was a degree of diversity in crop choice and farming strategies in different parts of the Indus Civilization across this time span (Petrie et al., 2016; Petrie and Bates, 2017; Weber, 1999; Weber et al., 2010). Farmers in southerly regions appear to have focused on summer or winter crops, while the more northern regions of Pakistan Punjab and Indian Punjab and Haryana were capable of supporting combinations of winter and summer crops (Petrie and Bates, 2017). Although there is evidence for diverse cropping practices involving both summer and winter crops in the northern areas during the urban period, agricultural strategies appeared to favor more intensive use of drought-resistant summer crops in the Late Harappan period (Madella and Fuller, 2006; Petrie and Bates, 2017; Pokharia et al., 2017; Weber, 2003; Wright, 2010). It has previously been suggested that weakened ISM was a major factor in these shifts (e.g. Giosan et al., 2012; Madella and Fuller, 2006). Based on our reconstruction of decreased IWM in northwest South Asia after 4.3 ka BP with a step-shift at 4.1 ka BP, we suggest that both IWM and ISM climatic factors played a role in shaping the human landscape. This includes the redistribution of population to smaller settlements in eastern regions with more direct summer rain, as well as the shift to increased summer crop dominated cropping strategies.

**6. Conclusion**

This study expanded on the $\delta^{18}O$ record of planktonic foraminifer in core 63KA of the northeastern Arabian Sea, originally published by Staubwasser et al. (2003). Using $\delta^{18}O$ of the surface-dwelling foraminifera *G. ruber*, the original study inferred an abrupt reduction in Indus River discharge at ~4.2 ka BP. Our further $\delta^{18}O$ analysis of a larger size fraction of this species corroborates maximum salinity at 4.1 and 3.95 ka BP. In addition, the $\delta^{18}O$ difference between the surface-dwelling *G. ruber* and slightly deeper-dwelling *G. sacculifer* ($\Delta\delta^{18}O_{s-r}$) reveals that surface waters were more saline than average for the period from 4.8-3.9 ka BP.

By also measuring a thermocline-dwelling planktonic foraminiferal species, *N. dutertrei*, we
infer an increase in the strength of the IWM between 4.5 and 4.3 ka BP, followed by reduction
in IWM-driven mixing that reaches a minimum at 4.1 ka BP.
Assuming that weaker IWM mixing implies a reduction in IWM rainfall amount or duration
over northwest South Asia under past climatic conditions, the 63KA core is used to infer
important changes in seasonal hydrology of the Indus River catchment. We propose that a
combined weakening of the IWM and ISM at 4.1 ka BP led to what has been termed the "4.2
ka BP" drought over northwest South Asia. The intersection of both a gradually weakening
ISM since 4.8 ka BP and a maximum decrease in IWM strength at 4.1 ka BP resulted in a
spatially layered and heterogeneous drought over a seasonal to annual timescale. Regions in
the western part of the Indus River basin accustomed to relying mainly on winter rainfall (also
via river run-off) would have been most severely affected by such changes. Regions in the
northeastern and eastern extents benefitted more from summer rainfall, and would have
been less severely affected, particularly as the ISM appears to recover strength by 3.9 ka BP.
Relatively strengthened IWM surface water mixing between 4.5 and 4.3 ka BP correlates with
a period of higher precipitation recorded at Karsandi on the northern margin of the Thar
Desert (Dixit et al., 2018), an area within the summer rainfall zone that is also sensitive to
small changes in winter precipitation. This time span also represents the beginnings of the
Mature Harappan phase (Possehl, 2002; Wright, 2010), which implies that increasingly
urbanized settlements may have flourished under a strengthened IWM. With a weakening of
the IWM at ~4.1 ka, eastern regions with more access to ISM rainfall may have been more
favorable locations for agriculture. This may also help explain the broad shift in population
towards more rural settlements in the northeastern extent of the Indus Civilization that
occurred by ~3.9 ka BP (Possehl, 1997; Petrie et al., 2017), and a shift to more drought-
tolerant kharif (summer) season crops in Gujarat (Pokharia et al., 2017) and at Harappa
(Madella and Fuller, 2006; Weber, 2003).
Given the importance of the relationships between humans and the environment during the
time of the Indus Civilization, understanding the impact of the IWM on precipitation
variability in northwest South Asia remains a critical area of research. We especially need a
better understanding of the wind patterns and moisture pathways that controlled the IWM
in the past. Disentangling both the length and intensity of seasonal precipitation is a crucial
aspect of understanding the impact of climate change on past societies, particularly in a
diverse region relying on mixed water sources (e.g., fluvial, ground aquifer, direct rainfall).
**Data availability**
Data presented in the paper can be accessed by contacting the corresponding author or
online at http://eprints.esc.cam.ac.uk/id/eprint/4371.
**Author contributions**
M.S. supplied core 63KA material, A.G. prepared the material for isotopic measurements, and
A.G. and D.A.H. interpreted the results. A.G., D.A.H., and C.A.P. wrote the manuscript.

**Competing interests**
The authors declare that they have no conflict of interest.
**Acknowledgements**
This research was carried out as part of the *TwoRains* project, which is supported by funding
from the European Research Council (ERC) under the European Union's Horizon 2020
research and innovation programme (grant agreement no 648609). The authors thank the
following persons at the University of Cambridge: Maryline Vautravers for foraminifera
identification, James Rolfe and John Nicolson for $\delta^{18}O$ measurements. We also thank our
editor and reviewers for comments that improved the manuscript.

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

## Supplemental figures and tables

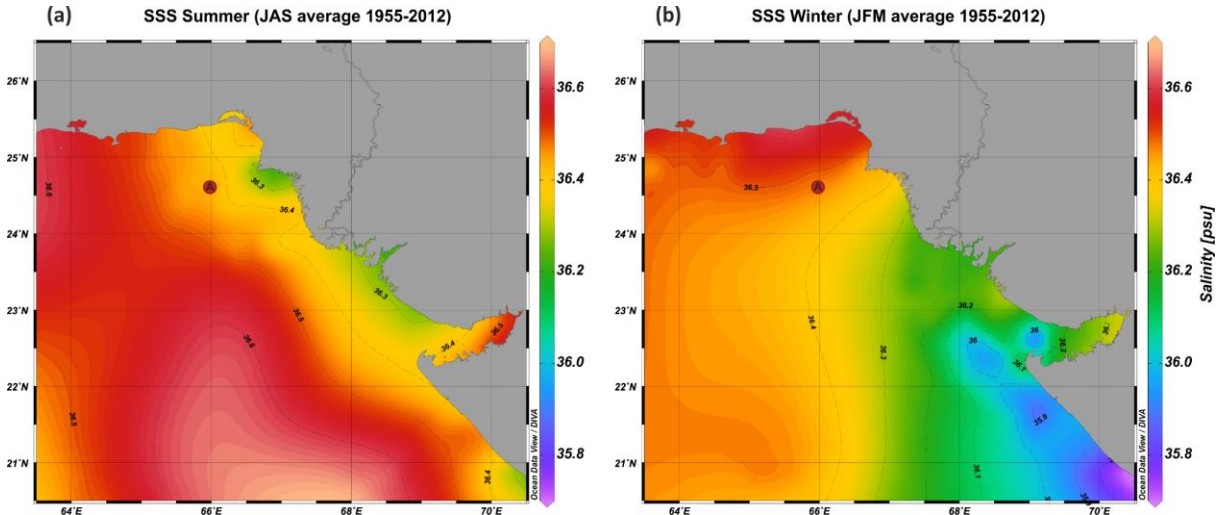

**Figure S1.** Mean surface salinity for 1955-2012, with data from the 2013 World Ocean Atlas (WOA) at 0.25° resolution (Zweng et al., 2013). Salinity contours are shown for **a.** summer (JAS) and **b.** winter (JFM). The Indus River is outlined. Note that over the time window of this dataset, modern Indus River discharge has been reduced by >50% due to barrages and irrigation (Ahmad et al., 2001). Plots created with Ocean Data Viewer (ODV).

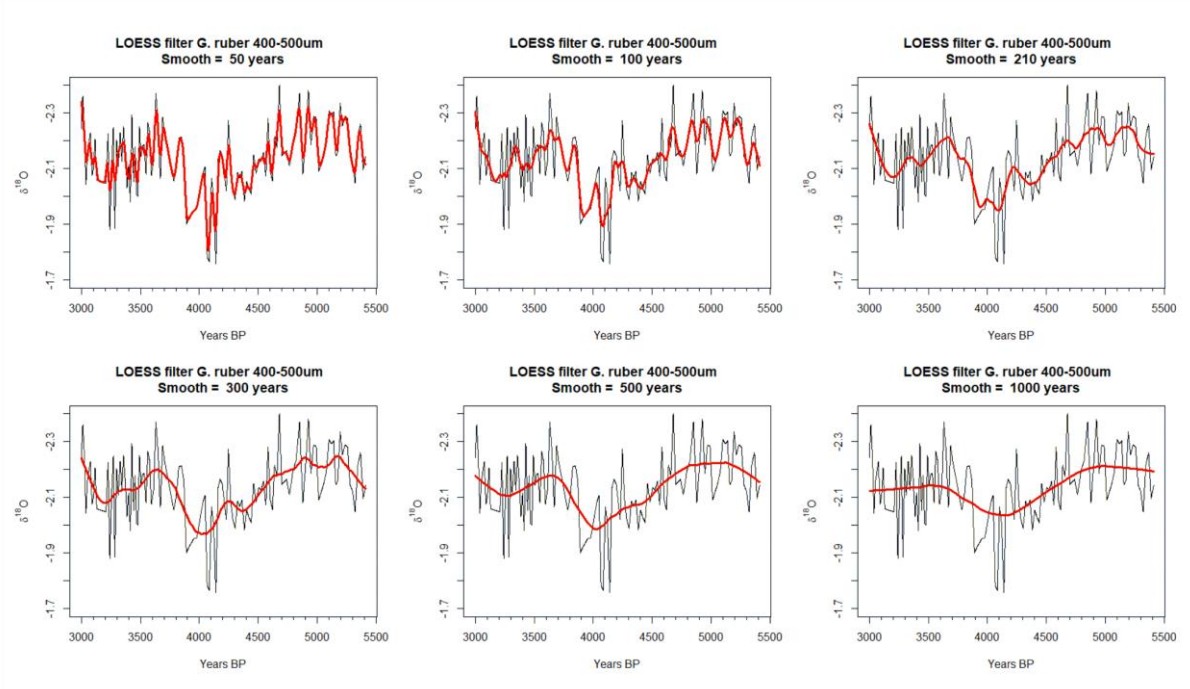

**Figure S2.** Comparison of loess smoothing windows of 50, 100, 210, 300, 500, and 1000 years for *G. ruber* in the 400-500µm fraction.

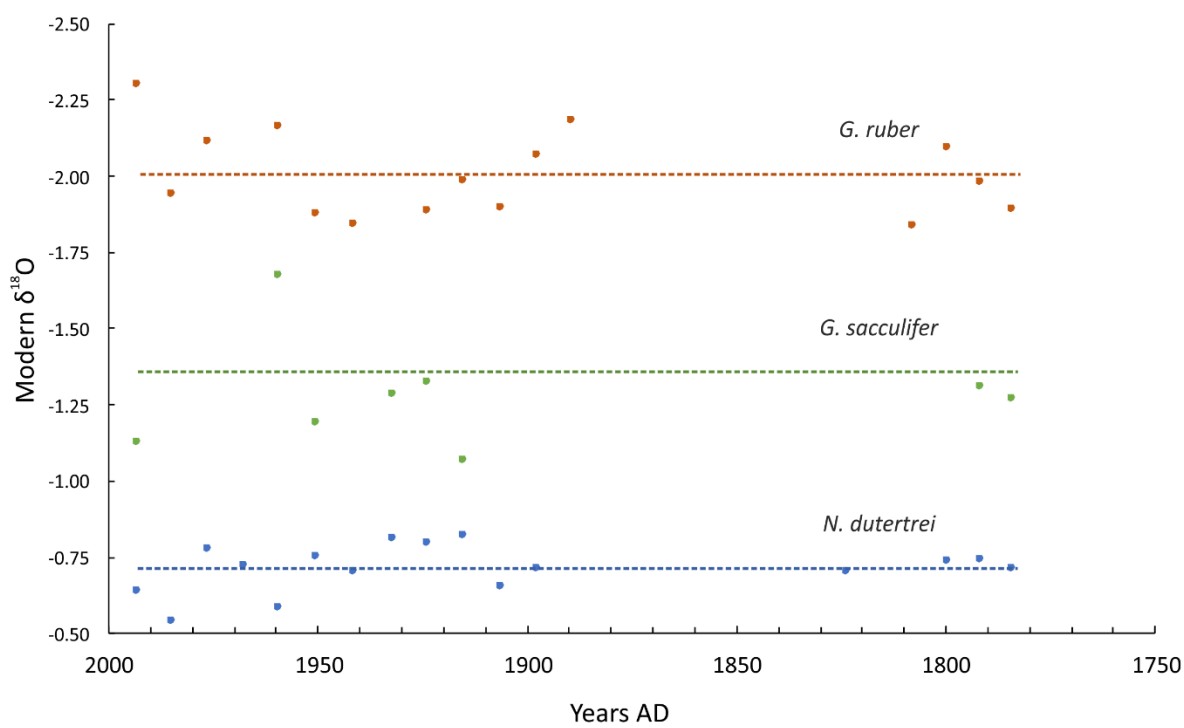

**Figure S3**. Modern $\delta^{18}O$ values of calcite, spanning approximately the last 200 years, measured from
surface sediment samples for all three species at the size fractions 315-400µm. Averages values for
the last 200 years (~1780-1993 AD) are compared to the period 5.4-3.0 ka BP: -2.01‰ (modern) and
-1.90‰ (old) for *G. ruber* (orange), -1.28‰ (modern) and -1.31‰ (old) for *G. sacculifer* (green), and -
0.72‰ (modern) and -0.76‰ (old) for *N. dutertrei* (blue).

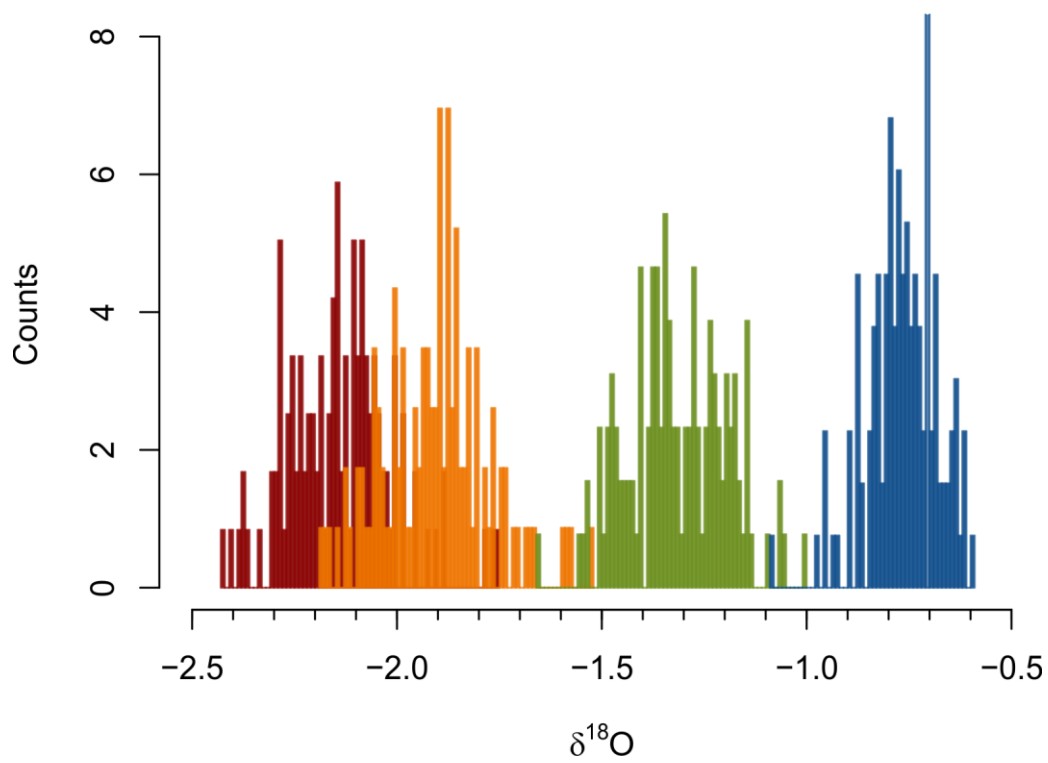

**Figure S4.** Frequency distributions of $\delta^{18}O$ data during 5.4-3.0 ka BP for *G. ruber* 400-500µm (red), *G.*
*ruber* 315-400µm (orange), *G. sacculifer* 315-400µm (green), *N. dutertrei* 315-400µm (blue).

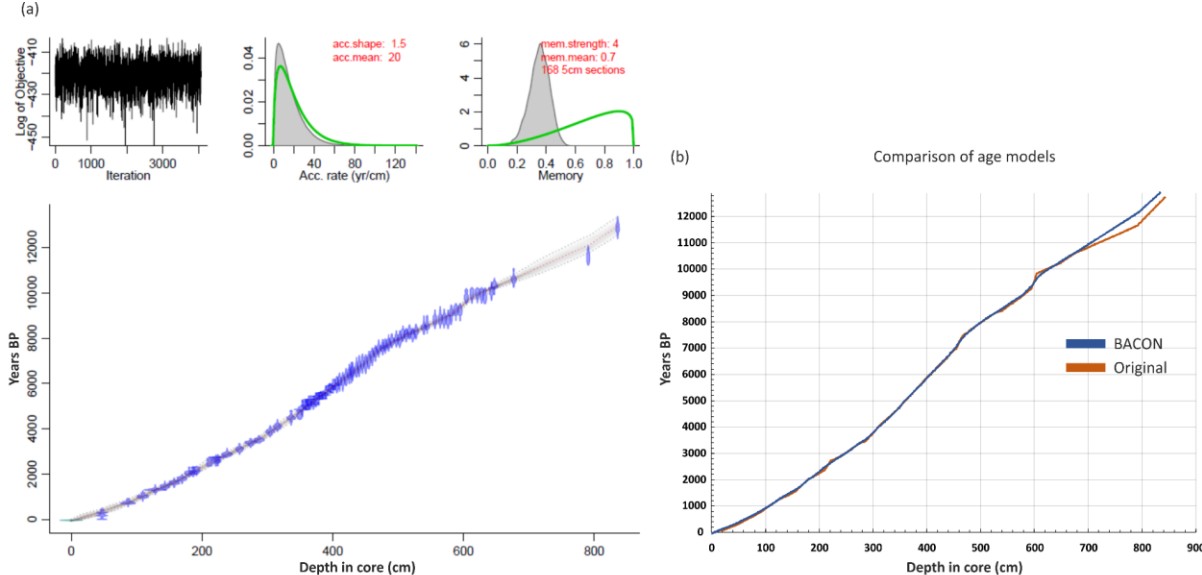

**Figure S5. a.** BACON age-depth model with calibrated dates shown in blue **b.** Age-depth model
comparison with the original published age model from Staubwasser et al. (2003) (orange) and the
new age model based on BACON software (blue).

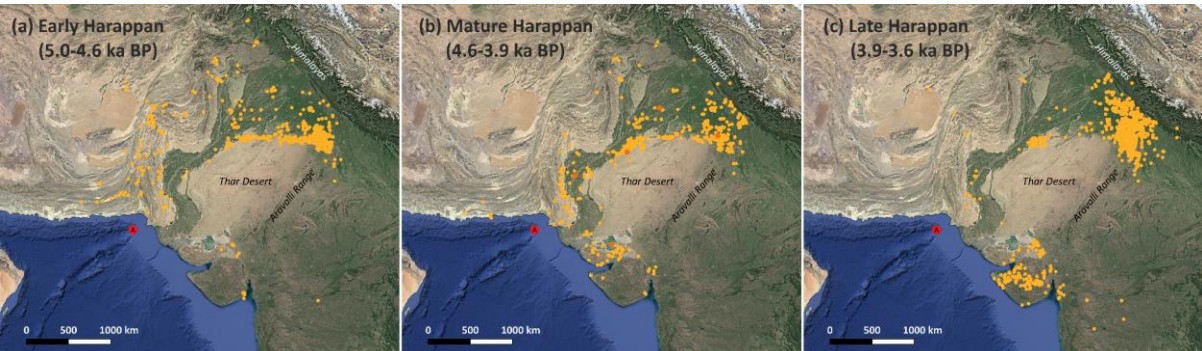

**Figure S6.** Indus site distributions (yellow points) during the **a.** Early Harappan (~5.0-4.6 ka BP), **b.**
Mature Harappan (~4.6-3.9 ka BP), and **c.** Late Harappan (~3.9-3.6 ka BP). Orange sites show larger
Harappan cities during the Mature Harappan period (Dholavira, Mohenjo Daro, Ganweriwala,
Harappa, and Rakhigarhi from bottom to top), core 63KA shown by red circle, background terrain from
Google Earth.
**Table S1.** Main statistical parameters of the $\delta^{18}$O data.

| | *G. ruber* 400-500µm | *G. ruber* 315-400µm | *G. sacculifer* 315-400µm | *N. dutertrei* 315-400µm |
|---|---|---|---|---|
| *n* | 119 | 115 | 129 | 132 |
| *Minimum* | -2.423 | -2.190 | -1.660 | -1.090 |
| *Maximum* | -1.752 | -1.520 | -1.000 | -0.590 |
| *1st Quartile* | -2.232 | -1.995 | -1.400 | -0.810 |
| *3rd Quartile* | -2.068 | -1.830 | -1.220 | -0.700 |
| *Mean* | -2.139 | -1.901 | -1.312 | -0.761 |
| *Median* | -2.144 | -1.890 | -1.320 | -0.760 |
| *Sum* | -254.58 | -218.66 | -169.26 | -100.46 |
| *SE Mean* | 0.012 | 0.012 | 0.011 | 0.007 |
| *LCL Mean* | -2.163 | -1.926 | -1.333 | -0.776 |
| *UCL Mean* | -2.116 | -1.877 | -1.291 | -0.746 |
| *Variance* | 0.016 | 0.017 | 0.015 | 0.007 |
| *Stdev* | 0.128 | 0.131 | 0.122 | 0.085 |

| | | | | |
|---|---|---|---|---|
| *Skewness* | 0.408 | 0.288 | -0.011 | -0.592 |
| *Kurtosis* | 0.511 | 0.174 | -0.364 | 0.850 |

**Table S2.** Age-Model calibration with BACON software.

| Depth (cm) | ¹⁴C date | Error (±1σ) | Reservoir (years) | IntCal13 min age BP | IntCal13 max age BP | IntCal13 mean age BP |
|---|---|---|---|---|---|---|
| *surface* | - | - | - | - | - | -43 |
| *47* | 790 | 30 | 565 | 267 | 309 | 288 |
| *87* | 1370 | 35 | 565 | 678 | 780 | 729 |
| *109.5* | 1665 | 30 | 565 | 952 | 1062 | 1007 |
| *128.5* | 1955 | 25 | 565 | 1283 | 1339 | 1311 |
| *143.5* | 2115 | 35 | 565 | 1369 | 1529 | 1449 |
| *157.5* | 2270 | 25 | 565 | 1552 | 1634 | 1593 |
| *169.5* | 2430 | 25 | 565 | 1728 | 1869 | 1799 |
| *180.5* | 2640 | 25 | 565 | 1988 | 2122 | 2055 |
| *186.5* | 2675 | 35 | 565 | 1993 | 2154 | 2074 |
| *191.5* | 2720 | 30 | 565 | 2044 | 2184 | 2114 |
| *211.5* | 3000 | 35 | 565 | 2356 | 2541 | 2449 |
| *221.5* | 3110 | 40 | 565 | 2491 | 2602 | 2547 |
| *224.5* | 3145 | 25 | 565 | 2708 | 2758 | 2733 |
| *238.5* | 3340 | 25 | 565 | 2836 | 2929 | 2883 |
| *257.5* | 3510 | 30 | 565 | 2999 | 3181 | 3090 |
| *274.5* | 3730 | 30 | 565 | 3343 | 3451 | 3397 |
| *287.5* | 3850 | 30 | 565 | 3450 | 3576 | 3513 |
| *304.5* | 4145 | 30 | 565 | 3828 | 3975 | 3902 |
| *315.5* | 4310 | 30 | 565 | 4062 | 4159 | 4111 |
| *336.5* | 4570 | 40 | 565 | 4408 | 4578 | 4493 |
| *349.5* | 4655 | 40 | 565 | 4512 | 4711 | 4612 |
| *353.5* | 4870 | 30 | 565 | 4832 | 4892 | 4862 |
| *357.5* | 5005 | 35 | 565 | 4952 | 5079 | 5016 |
| *360.5* | 4980 | 30 | 565 | 4868 | 5057 | 4963 |
| *363.5* | 5080 | 30 | 565 | 5050 | 5194 | 5122 |
| *366.5* | 5105 | 35 | 565 | 5053 | 5189 | 5121 |
| *370.5* | 5070 | 35 | 565 | 5046 | 5300 | 5173 |
| *374.5* | 5160 | 40 | 565 | 5372 | 5463 | 5418 |
| *378.5* | 5210 | 40 | 565 | 5303 | 5469 | 5386 |
| *381.5* | 5315 | 30 | 565 | 5460 | 5585 | 5523 |
| *385.5* | 5315 | 35 | 565 | 5453 | 5586 | 5520 |
| *389.5* | 5420 | 35 | 565 | 5580 | 5654 | 5617 |
| *395.5* | 5635 | 35 | 565 | 5741 | 5907 | 5824 |
| *398.5* | 5610 | 35 | 565 | 5713 | 5904 | 5809 |
| *402* | 5750 | 40 | 565 | 5891 | 6008 | 5950 |
| *406.5* | 5830 | 35 | 638 | 5899 | 6002 | 5951 |
| *410.5* | 5965 | 40 | 638 | 5994 | 6210 | 6102 |
| *415.5* | 5980 | 45 | 638 | 5997 | 6216 | 6107 |
| *420.5* | 6120 | 45 | 638 | 6201 | 6351 | 6276 |

| | | | | | |
|---|---|---|---|---|---|
| *425.5* | 6265 | 45 | 638 | 6311 | 6490 | 6401 |
| *428.5* | 6335 | 55 | 638 | 6395 | 6639 | 6517 |
| *430.5* | 6345 | 60 | 638 | 6396 | 6657 | 6527 |
| *436.5* | 6440 | 40 | 638 | 6495 | 6678 | 6587 |
| *440.5* | 6540 | 55 | 638 | 6627 | 6883 | 6755 |
| *445.5* | 6665 | 45 | 638 | 6773 | 6984 | 6879 |
| *450.5* | 6650 | 40 | 638 | 6749 | 6948 | 6849 |
| *455.5* | 6960 | 45 | 824 | 6912 | 7162 | 7037 |
| *460.5* | 7155 | 45 | 824 | 7166 | 7331 | 7249 |
| *465.5* | 7310 | 45 | 824 | 7308 | 7480 | 7394 |
| *470.5* | 7480 | 55 | 824 | 7438 | 7606 | 7522 |
| *476.5* | 7550 | 50 | 824 | 7551 | 7670 | 7611 |
| *480.5* | 7815 | 55 | 1011 | 7571 | 7743 | 7657 |
| *485.5* | 7920 | 70 | 1011 | 7617 | 7867 | 7742 |
| *490.5* | 8070 | 50 | 1011 | 7788 | 7976 | 7882 |
| *497.5* | 8130 | 55 | 1011 | 7837 | 8027 | 7932 |
| *502.5* | 8115 | 55 | 1011 | 7828 | 8020 | 7924 |
| *507.5* | 8400 | 60 | 1011 | 8148 | 8345 | 8247 |
| *512.5* | 8350 | 50 | 1011 | 8020 | 8218 | 8119 |
| *517.5* | 8490 | 50 | 1011 | 8194 | 8381 | 8288 |
| *522.5* | 8355 | 60 | 1011 | 8023 | 8312 | 8168 |
| *527.5* | 8510 | 60 | 1011 | 8194 | 8400 | 8297 |
| *539.5* | 8790 | 60 | 1118 | 8384 | 8563 | 8474 |
| *544.5* | 8880 | 55 | 1118 | 8425 | 8631 | 8528 |
| *556.5* | 9060 | 50 | 1118 | 8637 | 8986 | 8812 |
| *564.5* | 9120 | 70 | 1118 | 8636 | 9026 | 8831 |
| *570.5* | 9110 | 50 | 1118 | 8698 | 9007 | 8853 |
| *576.5* | 9060 | 50 | 1118 | 8637 | 8986 | 8812 |
| *581.5* | 9260 | 50 | 1118 | 8999 | 9153 | 9076 |
| *588.5* | 9390 | 50 | 1118 | 9119 | 9430 | 9275 |
| *595* | 9370 | 60 | 1118 | 9076 | 9419 | 9248 |
| *604.5* | 9570 | 50 | 781 | 9602 | 9952 | 9777 |
| *613* | 9660 | 70 | 781 | 9736 | 10194 | 9965 |
| *621.5* | 9670 | 50 | 781 | 9884 | 10189 | 10037 |
| *628* | 9650 | 70 | 781 | 9732 | 10188 | 9960 |
| *633* | 9570 | 80 | 781 | 9581 | 9963 | 9772 |
| *643* | 9770 | 70 | 781 | 9906 | 10251 | 10079 |
| *647.5* | 9920 | 60 | 781 | 10206 | 10436 | 10321 |
| *677* | 10160 | 60 | 781 | 10480 | 10752 | 10616 |
| *791* | 11145 | 50 | 1095 | 11325 | 11806 | 11566 |
| *836* | 12285 | 55 | 1300 | 12726 | 12995 | 12861 |