# Peer review of "Indian winter and summer monsoon strength over the 4.2 ka BP event in foraminifer isotope records from the Indus River delta in the Arabian Sea"

_Climate of the Past, 2018_

## Referee Comment (RC1) · A. Sinha (Referee) · 1 Oct 2018

Staubwasser et al back in 2003 published an influential paper entitled 'Climate change at the 4.2 ka BP termination of the Indus valley civilization and Holocene south Asian monsoon variability'. The paper generated a lot of interest and if I am not mistaken, was the first attempt of its kind, to reconstruct variations in Indus river discharge, and thereby, the hydroclimatic history of the region, where the Indus valley civilization emerged, prospered and declined. Fast forward 15 years, Giesche et al manuscript "Re-examining the 4.2 ka BP event in foraminifer isotope records from the Indus River delta in the Arabian Sea" seeks to confirm and improve upon the results from the original study. A salient and an important aspect of this manuscript is that the authors have attempted to reconstruct changes in both summer and winter Indian monsoon by adding new data through the stable oxygen and carbon isotopes analyses of two additional foraminiferal species over a targeted interval from 3.4 to 5.0 ka. They have presented their findings in the context of societal changes that occurred in this region. Additionally, their new data is a step towards a better dynamical understanding of the 4.2 ka event in the Indian subcontinent. Both are critical and important objectives. The paper is well written but its presentation could be further improved (see below). Authors are suitably careful in interpreting the proxy data. The conclusions are reasonably well supported by data and the necessary caveats are clearly noted. I, therefore, highly recommend this manuscript to be published in Climate of the Past with some modest revisions.

I have outlined my suggestions below that I hope authors will find useful in further improving the 'scientific and presentation qualities' of their manuscript.

Manuscript Length: First, I encourage authors to substantially reduce the length of this manuscript. A shortened/concise manuscript will greatly improve its readability. While the authors can decide how to best approach this, I would suggest that figures # 4, 6, and perhaps 7 (and associated discussions) can be easily moved into the supplementary section. Also, aren't Figures # 3 and 5 somewhat redundant? Can they be merged as a single figure?

Figures Presentation: Figure 1: Fonts are too small particularly on panels b and c. Figure 2c has multiple shades for each species. Are different shades representing sediment trap data for different depths? It is not clear from the figure caption. Figures # 3, 5, 7, 9, and 10: Loess smoothed curves are shown along with the raw data (is that correct?). The latter is shown as scatter plots, which is fine if the goal is to show the effectiveness of Loess smoothing but not very intuitive if one wishes to see the evolution of higher frequency variability in time series. I suggest perhaps one figure with scatter plot can be shown to demonstrate the idea of Loess smoothing and the rest of the

figures can be more conventional (i.e., line plots) superimposed by smoothed curves.

Statistical Treatment I would also recommend that authors present foraminifera isotope data as z-scores or anomalies from the mean (that is of course, after initially presenting raw data). This is particularly useful as comparisons are made across the different species and proxy records.

Authors have reported the results of several statistical tests in this manuscript. It is not clear whether the statistics were performed on interpolated or raw data. This needs some clarification. What is the average temporal resolution of non-interpolated isotope data? A critical information that I could not find in the manuscript.

---

## Author Comment (AC1) · 1 Oct 2018

Please find all d18o and d13c data available in the attached file.

Please also note the supplement to this comment:
https://www.clim-past-discuss.net/cp-2018-104/cp-2018-104-AC1-supplement.zip

---

## Referee Comment (RC2) · Anonymous Referee #2 · 10 Oct 2018

Giesche, et al. "Re-examining the 4.2 ka BP event in foraminifer isotope records from the Indus River delta in the Arabian Sea

Summary:

A precise characterization of the timing, duration and climatic expression of the 4.2 ka event in the Indian subcontinent is crucial in order to assess its potential linkage with the transformation of the Indus valley civilization. Giesche et al. provide new foraminiferal isotope data from the northeast Arabian Sea that spans a critical interval of time ($\sim$ 3.3 to 5.3 ka)—an interval of considerable interest from both climatic and societal points of view. The new oxygen isotope data of surface-and-thermocline-dwelling foraminiferal

species is (ultimately) interpreted by the authors in the terms of temporal variations in the strength of Indian summer and winter monsoons, respectively. The results (and proxy interpretations) while complements the results from Staubwasser et al (2003) study, I think this manuscript provides an improved (and less-speculative) discussion of plausible climatic and societal linkages. The manuscript is fairly well-written (but on the verbose side), logically-structured, but the graphical presentation of figures requires some attention. I recommend this manuscript be accepted for publication in Climate of the Past after some revisions.

Specific Comments: I think the title "Re-examining the 4.2 ka. . ..Arabian Sea" would seem to suggest that the focus of this manuscript lies in the "re-examining" part but in reality, it constitutes a rather small component of this manuscript. While generally complementary, I think there are also important differences in the results between the two studies (see figure 3b), which may have important bearing in assessing the timing and abruptness of the 4.2 event. I encourage authors to find a title that is more representative of the main conclusions/results.

Text and other information on Figure 1 are hard to follow. A different color scheme would also help. Figures with scatter plots (with the exception of Figure # 3b) are also not particularly helpful. Why not show these figures like figure S2? There is also a fair bit of redundancy between sections 2, 5.1, 5.2, and 5.3. I think the manuscript will benefit from a more concise presentation. Line 134-35: Consider rephrasing " northeastern regions close to New Delhi" Line 141: "while many paleoclimate studies. . ." Provide some reference here. Line 180: Fleitmann et al 2007 is a tertiary reference. Remove it. Line 280-281: Provide a statement that explicitly compares the age models between Staubwasser 2003 and this study. I think there is no difference in age models between the two studies but it needs to be spelled out here. Also, indicate the temporal resolution of the new dataset. Line 333: "The bandwidth of 210 years was considered optimal time window for capturing the overall trends in the dataset". What constitutes optimal? Why 210 years? It seems like a large window to smooth data

to discern the onset and termination of the 4.2 event. Line 351-352: Please quantify "weak" and "strong" correlation and provide statistical significance. Line 415 and 421 : consider rephrasing "trend" with "change" or something similar. Line 432-433: Support this by a suitable reference and briefly explain why "differencing" will improve the signal-to-noise-ratio. Lines 556-565: Some of this should be in the "Introduction" and not here.

---

## Referee Comment (RC3) · Anonymous Referee #3 · 23 Oct 2018

The authors present fascinating new data from core 63KA from the Arabian Sea to reconstruct changes in Indian Summer Monsoon rainfall over the adjacent continent and Indian Winter Monsoon strength. Compared with the original work by Staubwasser et al. (2003) this study presents new d18O records from subsurface and thermocline-dwelling foraminifera species. The difference between subsurface and surface foram-d18O reflects the intensity of surface freshening, whereas the difference between subsurface and thermocline foram-d18O is a measure of wind-driven vertical mixing. The authors focus on the time period 5.3 to 2.9 ka BP encompassing the major shift in both summer and winter monsoons at ∼4.2 ka. This mid-Holocene climate change as seen in the 63KA records is compared with the numerous land and marine data that have

been published since Staubwasser et al.'s study. The interpretation of the new data is sound. Taken individually, each sub-section of the Results and Discussion is well written and clear.

The problem of the manuscript is that the study aims have not been sufficiently worked out. The authors provide an overview on the state of knowledge, but they should more clearly work out the problems and "missing pieces". Indicate possible solutions, and then describe your own approach (which exactly follows those "possible solutions"). This information must be more clearly and prominently provided in the Introduction and not postponed until the Discussion; otherwise, the reader has no guideline for following the manuscript. As it stands, the Abstract and Introduction present the manuscript as a replicate of Staubwasser et al. (2003) with some additional data. But actually these additional data (N. dutertrei and G. sacculifer records) and their interpretation make up the core and primary scientific asset of this study.

Specific comments

1. Abstract, lines 64-65: See above. Even though the G. sacculifer and N. dutertrei records provide the key data for this study, they are presented as by-products, and only in the following sentence (line 66) the reader is informed why they have been generated in the first place.

2. Introduction, lines 124-128: Be more specific on the importance of the IWM. Reconstructing the IWM is one main part of this study, and hence its significance should be sufficiently highlighted.

3. Lines 146-153: Some explanation on the new G. ruber record is required. I guess that the N. dutertrei and G. sacculifer samples are from different sampling positions than the G. ruber samples from Staubwasser et al., and a new G. ruber record is necessary for calculating Dd18O (ruber-sacculifer). This is fine, but should be mentioned.

4. Methods, line 317ff, and Fig. 2d: These CTD data are a snapshot from a single

day. I would prefer profiles from the World Ocean Atlas, as these are probably more representative. Provide temperature and salinity profiles for two seasons, one covering the main fluxes of G. ruber and G. sacculifer (July-September), the other the peak occurrence of N. dutertrei (December). This will also give the reader an idea on how much seasonality is present at different water depths.

5. Line 329: Provide the total number of samples and the average temporal resolution of the raw data.

6. Results, line 362 and throughout: What is the number of degrees of freedom when calculating the p-values, do the authors use the number of actually measured data or the number of annually interpolated data?

7. Discussion, line 454ff: "is confirmed" should be toned down. The authors are correct as far as the main conclusions of the study are concerned, but otherwise the two records are not congruent. Do different test sizes potentially reflect different seasons?

Minor points

8. Line 238: Down to 100 m.

9. Line 253: recording the d18O and temperature of the seawater

10. Line 256-258: Please, rephrase.

11. Fig. 2a: Use stronger color contrasts.

12. Line 504: Add small delta (same format as in subsequent sentence).

---

## Short Comment (SC1) · 2 Nov 2018

Comment on Giesche et al. by L. Giosan (Woods Hole Oceanographic Institution) and K. Thirumalai (Brown University):

Giesche et al. present a valuable new dataset of planktonic foraminifer isotopic time series from core 63KA in the Arabian Sea. The authors briefly mention our recent paper on a similar topic (Giosan et al., 2018, Climate of the Past, in press). Giesche et al. expand on the original study by Staubwasser et al. (2003), but similar to this previous work, the new data exhibit low signal-to-noise ratio in a very complex coastal region. We argue that a more conservative interpretation is required to take into account this.

[Figure]

Addressing uncertainties is needed to convincingly show that salinity signals, which are indicative of a "4.2 ka event", or any such millennial/centennial events in the late Holocene for that matter, are detectable in the foram 18O (and 13C) composition in this region.

Surface water masses in the NE Arabian Sea at core 63KA location may be affected by (a) advection of waters from NW Arabian Sea that have a variable upwelling-modified composition; (b) fluvial discharge from the Indus but also from River Hub that is proximal to the core (figure 1 in Giesche et al. supplementary materials); (c) changes in winter to summer rain and snow/ice meltwater with variable isotopic signal that feed the Indus; (d) deep winter mixing bringing Arabian Sea High Salinity Water Mass (ASWHSW) to the surface. All these potential sources and/or modifiers affect the isotopic signal in planktonic forams. For example, ASWHSW mixing would increase the salinity and decrease the temperature of surface waters.

The dynamics of these waters masses is also complex near the coast. For example, the effect of the Indus freshwater plume at the core location is uncertain as summer coastal circulation is directed in the opposite direction along the coast of India. In fact, this is obvious in the modern salinity map provided by the authors (figure 1 in Giesche et al. supplementary materials) where the change in signal at the core location is close to none between summer and winter (< 0.2 psu). If anything, River Hub discharge could affect the salinity at the core site more than the Indus (same figure).

Given this complexity, despite any statistical tests, we argue that interpreting a signal of 0.04-0.07‰ as "significant" or "weakly significant" when intra-sample standard deviation is on the order of 0.12‰ is misleading. Smoothing of the signal and ulterior correlation at a subjectively-chosen window is bound to produce some degree of significance even in random data. The fact that there is no significant correlation in sample-to-sample comparison of the same species (G. ruber) at different size fractions is unsettling and should be taken as a warning signal.

Are the proxies chosen by Giesche et al. appropriate in these conditions to the task of reconstructing the summer and winter monsoons? We argue that the authors do not make a convincing case for this. First, their indicator for winter mixing, N. dutertrei, does not preferentially live in winter. Assuming that the limited sediment trap data cited by the authors is correct, the summer peak abundance in N. dutertrei is as important quantitatively as the winter peak due to its more extended temporal range (4 months compared to 1-2 months in winter). Thus isotopic signals in this species will be a mixed summer-winter 18O and not appropriate for detecting a winter monsoon signal. Furthermore, interpretation of 13C values in this and other planktonic species is too simplistic given their known problems (e.g., possible shift in habitat, vital effects). Such problems are not discussed in the paper and interpretation is not even supported in the only cited reference (encyclopedia entry by Lynch-Stieglitz, 2006).

In these conditions it is not productive to extend further our analysis of the paper as all interpretation and conclusions are vitiated by inappropriate basic assumptions. We urge the authors to consider a more conservative approach in interpreting this new data. It is evident to us that solving the salinity signal using forams in this region needs a more sophisticated approach (e.g., Ba/Ca in planktonics; temperature correction from Mg/Ca measurements, etc.). The alkenone-based SST estimates from Doose-Rolinski et al. can only be used to understand a qualitative indicative range of cooling as we now know that the high temperature plateau of the alkenone method limits its usefulness given the high SSTs in the region.

---

## Author Comment (AC2) · 2 Dec 2018

*Staubwasser et al back in 2003 published an influential paper entitled 'Climate change at the 4.2 ka BP termination of the Indus valley civilization and Holocene south Asian monsoon variability'. The paper generated a lot of interest and if I am not mistaken, was the first attempt of its kind, to reconstruct variations in Indus river discharge, and thereby, the hydroclimatic history of the region, where the Indus valley civilization emerged, prospered and declined. Fast forward 15 years, Giesche et al manuscript "Re-examining the 4.2 ka BP event in foraminifer isotope records from the Indus River delta in the Arabian Sea" seeks to confirm and improve upon the results from the original study. A salient and an important aspect of this manuscript is that the authors have attempted to reconstruct changes in both summer and winter Indian monsoon by adding new data through the stable oxygen and carbon isotopes analyses of two additional foraminiferal species over a targeted interval from 3.4 to 5.0 ka. They have presented their findings in the context of societal changes that occurred in this region. Additionally, their new data is a step towards a better dynamical understanding of the 4.2 ka event in the Indian subcontinent. Both are critical and important objectives. The paper is well written but its presentation could be further improved (see below). Authors are suitably careful in interpreting the proxy data. The conclusions are reasonably well supported by data and the necessary caveats are clearly noted. I, therefore, highly recommend this manuscript to be published in Climate of the Past with some modest revisions.*

We thank Ashish Sinha for taking the time to review our manuscript and provide constructive feedback.

*I have outlined my suggestions below that I hope authors will find useful in further improving the 'scientific and presentation qualities' of their manuscript.*

*Manuscript Length: First, I encourage authors to substantially reduce the length of this manuscript. A shortened/concise manuscript will greatly improve its readability. While the authors can decide how to best approach this, I would suggest that figures # 4, 6, and perhaps 7 (and associated discussions) can be easily moved into the supplementary section. Also, aren't Figures # 3 and 5 somewhat redundant? Can they be merged as a single figure?*

We agree that some of the figures can be consolidated. We have now merged figures 3,5, and 7 (the resulting figure is now called figure 3). We have tried to make the language more precise throughout the manuscript to improve the readability, and have removed some redundancies. These changes can be viewed in the revised version of the manuscript, at the end of this file.

*Figures Presentation: Figure 1: Fonts are too small particularly on panels b and c. Figure 2c has multiple shades for each species. Are different shades representing sediment trap data for different depths? It is not clear from the figure caption.*

The font size has been enlarged for Figure 1. You are correct that the various shadings in Figure 2c show different trap data (deep v. shallow traps). This information has been added to the figure caption.

*Figures # 3, 5, 7, 9, and 10: Loess smoothed curves are shown along with the raw data (is that correct?). The latter is shown as scatter plots, which is fine if the goal is to show the effectiveness of Loess smoothing but not very intuitive if one wishes to see the evolution of higher frequency variability in time series. I suggest perhaps one figure with scatter plot can be shown to demonstrate the idea of Loess smoothing and the rest of the figures can be more conventional (i.e., line plots) superimposed by smoothed curves.*

We agree that line plots show important information about the raw data over time, therefore we have included lines for the plots in figure 3 (previously split into figures 3/5/7). We have also included lines for figures 7 & 8 (previously called 9 & 10).

*Statistical Treatment I would also recommend that authors present foraminifera isotope data as z-scores or anomalies from the mean (that is of course, after initially presenting raw data). This is particularly useful as comparisons are made across the different species and proxy records.*

We agree that z-scores can be a useful way of presenting the data and its variability in comparison to other proxies. Because our data and the comparisons are mainly based on $\delta^{18}O$, we think that the original units of the records are more informative for the reader. Instead, we have uploaded the raw data file, and readers are encouraged to re-plot the data as z-scores if it is useful for their purposes.

*Authors have reported the results of several statistical tests in this manuscript. It is not clear whether the statistics were performed on interpolated or raw data. This needs some clarification. What is the average temporal resolution of non-interpolated isotope data? A critical information that I could not find in the manuscript.*

We recognize that our original description of the statistical tests did not clarify these important points, and have improved the relevant sections. All statistical tests (t-tests, correlations, SiZer analysis) were performed on non-interpolated raw data (missing depths were ignored). We have now included "n" number of data points for the t-tests, and the p value for the correlation in the Results section. The loess smoothing was generated purely for visualization purposes, and we use a smoothing window/span equivalent to 210/2426 (where 210 represents the years of the smoothing window and 2426 represents the total number of years covered by the dataset). The average temporal resolution of the non-interpolated data is 18 years/cm, with a range of 12-29 years.

Revised manuscript below.

[revised manuscript text omitted]

---

## Author Comment (AC3) · 2 Dec 2018

Giesche, et al. "Re-examining the 4.2 ka BP event in foraminifer isotope records from the Indus River delta in the Arabian Sea

*Summary:*
*A precise characterization of the timing, duration and climatic expression of the 4.2 ka event in the Indian subcontinent is crucial in order to assess its potential linkage with the transformation of the Indus valley civilization. Giesche et al. provide new foraminiferal isotope data from the northeast Arabian Sea that spans a critical interval of time (~ 3.3 to 5.3 ka), an interval of considerable interest from both climatic and societal points of view. The new oxygen isotope data of surface-and-thermocline-dwelling foraminiferal species is (ultimately) interpreted by the authors in the terms of temporal variations in the strength of Indian summer and winter monsoons, respectively. The results (and proxy interpretations) while complements the results from Staubwasser et al (2003) study, I think this manuscript provides an improved (and less-speculative) discussion of plausible climatic and societal linkages. The manuscript is fairly well-written (but on the verbose side), logically-structured, but the graphical presentation of figures requires some attention. I recommend this manuscript be accepted for publication in Climate of the Past after some revisions.*

We thank reviewer #2 for taking the time to provide a review with helpful comments on our manuscript.

*Specific Comments: I think the title "Re-examining the 4.2 ka. . ..Arabian Sea" would seem to suggest that the focus of this manuscript lies in the "re-examining" part but in reality, it constitutes a rather small component of this manuscript. While generally complementary, I think there are also important differences in the results between the two studies (see figure 3b), which may have important bearing in assessing the timing and abruptness of the 4.2 event. I encourage authors to find a title that is more representative of the main conclusions/results.*

Both reviewers 2&3 have addressed this point in their comments. We agree with this assessment, and have proposed a new title for the revised manuscript: "**Indian winter and summer monsoon strength over the 4.2 ka BP event in foraminifer isotope records from the Indus River delta in the Arabian Sea**"

*Text and other information on Figure 1 are hard to follow. A different color scheme would also help.*

We have edited Figure 1 to increase font size where possible, as well as increased the contrast between colors of the text and background. These changes can be viewed in the revised version of the manuscript, at the end of this file.

*Figures with scatter plots (with the exception of Figure # 3b) are also not particularly helpful. Why not show these figures like figure S2?*

We have added lines between data points in Figures 3/5/7, which have been consolidated into a single figure (now called Figure 3) following the comments of Reviewer #1. We have also added lines between data points in Figures 7 & 8 (previously called 9 & 10).

*There is also a fair bit of redundancy between sections 2, 5.1, 5.2, and 5.3. I think the manuscript will benefit from a more concise presentation.*

We have now reduced redundancies by only mentioning elements of site description in section 2. For example, we removed a paragraph at the start of section 5.3, and removed or edited several sentences in section 5.1.

*Line 134-35: Consider rephrasing "northeastern regions close to New Delhi"*

Done.

*Line 141: "while many paleoclimate studies…" Provide some reference here.*

Done.

*Line 180: Fleitmann et al 2007 is a tertiary reference. Remove it.*

Done.

*Line 280-281: Provide a statement that explicitly compares the age models between Staubwasser 2003 and this study. I think there is no difference in age models between the two studies but it needs to be spelled out here. Also, indicate the temporal resolution of the new dataset.*

A new age model was made using updated (Bayesian) modeling software and updated radiocarbon calibrations (IntCal13), but there is no major difference between the age models except during the period 13-11 ka BP, which is out of the range of interest for this paper (a visual comparison of age models can be seen in supplemental figure S5b). The temporal resolution of the interval of interest (18 years/cm) has now been noted in two places - Section 2.2 and 3.1.

*Line 333: "The bandwidth of 210 years was considered optimal time window for capturing the overall trends in the dataset". What constitutes optimal? Why 210 years? It seems like a large window to smooth data to discern the onset and termination of the 4.2 event.*

We trialed several smoothing windows for the data, which can be viewed in Supplemental figure S2. We agree that "optimal" is subjective and rephrased it to "reasonable". 210 years were picked for several reasons: 1) It was the original smoothing window applied to the dataset in Staubwasser et al., 2003. 2) This represents the solar Gleissberg cycle of ~210 years, and helps smooth out any influence of the century-scale solar cycles if present. 3) Considering the relatively low signal to noise ratio of the record, we wanted to ensure that we were not exaggerating the significance of only a few data points – 210 years incorporates ~12 cm of the record, and the resulting smoothing shows the most obvious trends in the data considering the variability. However, as we have now added lines between the raw data points in figures 3, 7 and 8, the reader is better able to assess the data independently of its smoothing.

*Line 351-352: Please quantify "weak" and "strong" correlation and provide statistical significance.*

We no longer report the correlation for smoothed *G. ruber* data (possible autocorrelation), but have included the p value for the Pearson's correlation for the raw data: weak positive correlation (R = 0.25, p < 0.01, n = 109, slope 0.26, intercept -1.36).

*Line 415 and 421 : consider rephrasing "trend" with "change" or something similar.*

Done.

*Line 432-433: Support this by a suitable reference and briefly explain why "differencing" will improve the signal-to-noise-ratio.*

We understand how this phrasing might be misleading, because differencing will not reduce the errors associated with each individual measurement. However, differencing can help emphasize signals of interest. If we wish to isolate the freshwater plume salinity signal, differences may remove the varying influence of other factors such as water temperature (without requiring an actual reconstruction of the temperature). While temperature changes would affect the surface dwellers *G. sacculifer* and *G. ruber* equally, the surface freshwater plume would primarily affect *G. ruber*. The advantages of differencing are discussed in more detail by Tian et al. (2005) and Ravelo and Shackleton (1995). We have rephrased this sentence in the manuscript to better reflect the intended meaning.

*Lines 556-565: Some of this should be in the "Introduction" and not here.*

We have removed this paragraph entirely from the section as the same information is provided in Section 2.2.

References mentioned in this response:

[revised manuscript text omitted]
 extend from the surface downup to 100 m depth (Joseph and Freeland, 2005). Theis high salinity can beis explained
by the high evaporative rates over this region. ASHSW forms in the winter, but is prevented
from reaching our coring site on the shelf by northerly subsurface currents until the summer
(Kumar and Prasad, 1999). Along coastal areas, the ASHSW is starkly contrasted by the fresh
water discharge of the Indus River, combined with direct precipitation. In contrast, surface
waters in the Bay of Bengal on the eastern side of India have much lower surface water
salinity, because of overall higher precipitation and stronger stratification from weaker winds
(Shenoi et al., 2002). The heightened evaporative conditions and highly saline surface waters
of the northeastern Arabian Sea make it a sensitive study location to observe changes in
discharge of the entire Indus River catchment area – ultimately tracking changes in monsoon
strength. Unlike individual terrestrial records, which may be affected by local climatic
processes, the marine record from core 63KA is more likely to integrate regional changes of
the large-scale ocean-atmosphere system.
Planktonic foraminifera complete their life cycle within a few weeks (Bé and Hutson, 1977).
Peak abundances indicate the time of year when each species tends to calcify, thereby
recording the $\delta^{18}O$ and temperature of the seawater in their $CaCO_3$ shells primarily during
certain seasons. Foraminifer abundances in the eastern Arabian Sea have been studied by
Curry et al. (1992) using sediment traps deployed at shallow (~1400 m) and deep (~2800 m)
water depths ("T" in Figure 1a). Peak abundances for G. ruber and G. sacculifer occur have
peak abundances during the summer months (June-September), whereas N. dutertrei peak
lives mainly during the winter as well asand has a secondary peak in with a secondary peak

[revised manuscript text omitted]

---

## Author Comment (AC4) · 2 Dec 2018

*The authors present fascinating new data from core 63KA from the Arabian Sea to re- construct changes in Indian Summer Monsoon rainfall over the adjacent continent and Indian Winter Monsoon strength. Compared with the original work by Staubwasser et al. (2003) this study presents new d18O records from subsurface and thermocline- dwelling foraminifera species. The difference between subsurface and surface foram- d18O reflects the intensity of surface freshening, whereas the difference between sub- surface and thermocline foram-d18O is a measure of wind-driven vertical mixing. The authors focus on the time period 5.3 to 2.9 ka BP encompassing the major shift in both summer and winter monsoons at ~4.2 ka. This mid-Holocene climate change as seen in the 63KA records is compared with the numerous land and marine data that have been published since Staubwasser et al.'s study. The interpretation of the new data is sound. Taken individually, each sub- section of the Results and Discussion is well written and clear.*

We thank reviewer 3 for their thoughtful evaluation and comments on our manuscript.

*The problem of the manuscript is that the study aims have not been sufficiently worked out. The authors provide an overview on the state of knowledge, but they should more clearly work out the problems and "missing pieces". Indicate possible solutions, and then describe your own approach (which exactly follows those "possible solutions"). This information must be more clearly and prominently provided in the Introduction and not postponed until the Discussion; otherwise, the reader has no guideline for following the manuscript. As it stands, the Abstract and Introduction present the manuscript as a replicate of Staubwasser et al. (2003) with some additional data. But actually these additional data (N. dutertrei and G. sacculifer records) and their interpretation make up the core and primary scientific asset of this study.*

The Abstract and Introduction have been edited to clarify the overall study aims. These changes can be viewed in the revised version of the manuscript, at the end of this file. We agree that our new findings were somewhat obscured under the title that implies we are only "re-examining" old data. Based on the comments by both reviewers 2 & 3, we have decided to propose a new title for the manuscript: "**Indian winter and summer monsoon strength over the 4.2 ka BP event in foraminifer isotope records from the Indus River delta in the Arabian Sea**"

*Specific comments*
*1. Abstract, lines 64-65: See above. Even though the G. sacculifer and N. dutertrei records provide the key data for this study, they are presented as by-products, and only in the following sentence (line 66) the reader is informed why they have been generated in the first place.*

We have now rearranged the order of introducing these datasets in the abstract to make clear that the new *G. sacculifer* and *N. dutertrei* records are the focus of the paper. Additionally, the revised title of the paper better describes the new findings of the research.

*2. Introduction, lines 124-128: Be more specific on the importance of the IWM. Reconstructing the IWM is one main part of this study, and hence its significance should be sufficiently highlighted.*

We have added two sentences in this paragraph to direct the focus to the IWM, as well as a more detailed introduction of the IWM proxy at the end of the Introduction.

*3. Lines 146-153: Some explanation on the new G. ruber record is required. I guess that the N. dutertrei and G. sacculifer samples are from different sampling positions than the G. ruber samples from Staubwasser et al., and a new G. ruber record is necessary for calculating Dd18O (ruber-sacculifer). This is fine, but should be mentioned.*

All foraminifera including the new data are from the exact same core, depths, sub-samples reported by Staubwasser et al. (2003). However, the picking of the two size fractions of *G. ruber* foraminifera and their geochemical analysis was done by different people ~15 years apart. Differences for *G. sacculifer - G. ruber* are reported for both size fractions of *G. ruber*. The main reason for measuring the *G. ruber* record was to replicate this dataset and assess if the salinity signal could be distinguished from other variability affecting the oxygen isotopes of this species. The correlation of the raw data, as well as the overall agreement of the long-term trends in the two independently measured records, supports the reproducibility of the data sets despite a low signal to noise ratio.

*4. Methods, line 317ff, and Fig. 2d: These CTD data are a snapshot from a single day. I would prefer profiles from the World Ocean Atlas, as these are probably more representative. Provide temperature and salinity profiles for two seasons, one covering the main fluxes of G. ruber and G. sacculifer (July-September), the other the peak occurrence of N. dutertrei (December). This will also give the reader an idea on how much seasonality is present at different water depths.*

We have now plotted WOA data covering both summer (JAS) and winter (JFM) seasons, and show these in Figure 2. We continue to use the September 1993 CTD profile from the PAKOMIN cruise for the equilibrium calcite calculations (Figure 5), because these data come from one set of measurements directly from the coring location.

*5. Line 329: Provide the total number of samples and the average temporal resolution of the raw data.*

The total number of depths analyzed was 132 (now added to the Results section). The average temporal resolution (18 years/cm) is given in Section 2.2 and 3.1 (now repeated in the manuscript, because this is crucial information and all reviewers overlooked this from line 222 in the original manuscript). For *N. dutertrei*, we obtained data for all depths, but for *G. sacculifer* there were insufficient foraminifera (gaps) for 3 depths plus one outlier, and for *G. ruber* (400-500μm) we had 14 gaps (one of which is an outlier), and for *G. ruber* (315-400μm) there were 17 gaps. This is noted in section 3.2.

*6. Results, line 362 and throughout: What is the number of degrees of freedom when calculating the p-values, do the authors use the number of actually measured data or the number of annually interpolated data?*

The number "n" of data points for all t-tests and correlations are now included in the manuscript. All statistical tests were performed on raw data, and interpolated data are generated only for the visualization of the 210-year smoothing in the plots.

*7. Discussion, line 454ff: "is confirmed" should be toned down. The authors are correct as far as the main conclusions of the study are concerned, but otherwise the two records are not congruent. Do different test sizes potentially reflect different seasons?*

We have now used the word "reflected" and point out the differences visible between both records. Importantly, the increase in $\delta^{18}O$ of the larger *G. ruber* size fraction begins much earlier than 4.2 ka BP (rather around 4.8 ka BP), indicating that the summer monsoon and freshwater discharge may have started to weaken earlier than 4.2 ka BP. Additional *G. ruber* trap data from the region would be needed to answer the question about whether the offset between size fractions is due to seasonality (perhaps larger size fractions are biased to the warm season), preferred depth (perhaps larger size fractions live closer to the surface?), or other physical characteristics.

*Minor points*

*8. Line 238: Down to 100 m.*

Done.

*9. Line 253: recording the d18O and temperature of the seawater*

Done.

*10. Line 256-258: Please, rephrase.*

This sentence has been simplified.

*11. Fig. 2a: Use stronger color contrasts.*

Figure 2a now has more strongly contrasting blue shades of color.

*12. Line 504: Add small delta (same format as in subsequent sentence).*

Done.

Revised manuscript below.

[revised manuscript text omitted]

---

## Author Comment (AC5) · 2 Dec 2018

Comment on Giesche et al. by L. Giosan (Woods Hole Oceanographic Institution) and K. Thirumalai (Brown University):

*Giesche et al. present a valuable new dataset of planktonic foraminifer isotopic time series from core 63KA in the Arabian Sea. The authors briefly mention our recent paper on a similar topic (Giosan et al., 2018, Climate of the Past, in press). Giesche et al. expand on the original study by Staubwasser et al. (2003), but similar to this previous work, the new data exhibit low signal-to-noise ratio in a very complex coastal region. We argue that a more conservative interpretation is required to take into account this.*

We thank L. Giosan and K. Thirumalai for taking the time to write a comment for the discussion of this manuscript and challenging us to sharpen our arguments.

*Addressing uncertainties is needed to convincingly show that salinity signals, which are indicative of a "4.2 ka event", or any such millennial/centennial events in the late Holocene for that matter, are detectable in the foram 18O (and 13C) composition in this region.*
*Surface water masses in the NE Arabian Sea at core 63KA location may be affected by (a) advection of waters from NW Arabian Sea that have a variable upwelling-modified composition;*

We acknowledge that our site location can be affected by multiple water masses and processes, but the existing paleo-SST records from the region favor a salinity-based explanation for the *G. ruber* $\delta^{18}O$ record and difference between *G. sacculifer* and *G. ruber* ($\Delta\delta^{18}O_{s-r}$). As discussed in the original paper of Staubwasser et al. (2003), we point to the data from core M5-422 off the coast of N. Oman in the NW Arabian Sea (Cullen et al., 2000) showing a decrease in *G. ruber* $\delta^{18}O$ over the 4.2 ka BP event, indicating that SSTs are warmer rather than cooler over this time. If anything, higher SSTs would suppress the signal of increased salinity we note in the surface-dwelling foraminifera. Similarly, it argues against the influence of cold upwelling water, because M5-422 is located downstream between the zone of upwelling and core 63KA. Finally, the SSTs recorded in the nearby core 56KA from the NE Arabian Sea (Doose-Rolinski et al., 2001), also show that temperature does not greatly change the salinity signal over 4.2 ka BP, which was discussed and shown in Figure 1 a-c of Staubwasser (2012).

*(b) fluvial discharge from the Indus but also from River Hub that is proximal to the core (figure 1 in Giesche et al. supplementary materials);*

We agree that the Hub River may also contribute freshwater, but the Indus river discharge (>100 km³ per year before ~1950) is orders of magnitude greater than the Hub River (0.1 km³ per year) (Milliman et al., 1984). Although the Hub river may contribute sediment, the relative amount of freshwater discharge is much smaller than the Indus. The arrival of freshwater at the coring location during summer months can be seen in the supplemental Figure S1 of the manuscript – this large plume extends along the entire coastline. Additionally, the salinity maps provided in Figure S1 include an important caveat in their

caption – over the time window of this map (1955-2012), modern Indus River discharge has been reduced by >50% due to barrages and irrigation (Ahmad et al., 2001). This means that the freshwater plume seen in the summer map is artificially reduced compared to the discharge before 1955. We conclude that the Indus River freshwater discharge is the most significant factor influencing surface water salinity at our coring location during summer months when *G. ruber* and *G. sacculifer* have peak abundances.

*(c) changes in winter to summer rain and snow/ice meltwater with variable isotopic signal that feed the Indus;*

Isotope mass balance calculations suggest that the ratio of winter to summer rain or snow/ice/meltwater to direct runoff in the Indus river is unlikely to influence the isotopic signal of the Indus to the degree that would be needed to impact the signal of the foraminifera. The isotopic composition of the Indus River is -11.1‰$_{VSMOW}$ (Karim and Veizer, 2002) compared to the Arabian Sea surface waters of ~1‰$_{VSMOW}$ (LeGrande and Schmidt, 2006). The interannual $\delta^{18}$O variability of Indus river discharge, ranging <2‰ (Lambs et al., 2005), mixed throughout ~20 m depth in the coastal NE Arabian Sea region would have a minor impact the $\delta^{18}$O of the foraminifera (we estimate no more than ±0.05‰ for a ±1‰ change in river composition). In contrast, the increase in $\delta^{18}$O of *G. ruber* at 4.1 ka BP exceeds the mean value by +0.38‰.

*(d) deep winter mixing bringing Arabian Sea High Salinity Water Mass (ASWHSW) to the surface. All these potential sources and/or modifiers affect the isotopic signal in planktonic forams. For example, ASWHSW mixing would increase the salinity and decrease the temperature of surface waters.*

Arabian Sea High Salinity Water (ASHSW) can be an important factor in the region. ASHSW forms in the surface waters of the northern Arabian Sea during winter due to intensified evaporation and cooling (Kumar and Prasad, 1999). This is the source of the highly saline surface waters in this part of the Arabian Sea. Two relevant points emerge from Kumar and Prasad's (1999) analysis: first, there is a northward current along the west coast of India during winter months that initially prevents the spreading of the high-salinity water onto the shelf (our coring location), and second, the high salinity water is then pushed northeast in the summer by the ISM. In fact, this is the highly saline water that provides the crucial contrast to the primarily summertime freshwater discharge of the Indus River. Additionally, our difference proxies ($\Delta^{18}$O) monitor changes in the water column regardless of the water mass composition throughout time. Differencing reduces the effect of ASHSW because the surface and deep waters are equally affected, but a freshwater plume would have a much greater impact on the surface. For example, the difference between surface dwelling species *G. sacculifer* and *G. ruber* ($\Delta\delta^{18}$O$_{s-r}$) would reflect the relative impact of the summertime freshwater plume (affecting *G. ruber* more than *G. sacculifer*). Additionally, the amount of warmer surface water mixing deep in the water column during winter (when ASHSW does not reach the coring location) would be reflected in the absolute $\delta^{18}$O of the thermocline-dwelling *N. dutertrei*.

*The dynamics of these waters masses is also complex near the coast. For example, the effect of the Indus freshwater plume at the core location is uncertain as summer coastal circulation is directed in the opposite direction along the coast of India. In fact, this is obvious in the modern salinity map provided by the authors (figure 1 in Giesche et al. supplementary materials) where the change in*

*signal at the core location is close to none between summer and winter (< 0.2 psu). If anything, River Hub discharge could affect the salinity at the core site more than the Indus (same figure).*

See above response to point b).

*Given this complexity, despite any statistical tests, we argue that interpreting a signal of 0.04-0.07‰ as "significant" or "weakly significant" when intra-sample standard deviation is on the order of 0.12‰ is misleading. Smoothing of the signal and ulterior correlation at a subjectively-chosen window is bound to produce some degree of significance even in random data. The fact that there is no significant correlation in sample-to-sample comparison of the same species (G. ruber) at different size fractions is unsettling and should be taken as a warning signal.*

We agree that the statistics must be carefully interpreted, but they cannot be ignored. The statistical tests take into account the variability of populations within the dataset. The Welch's t-test comparing the mean values for *N. dutertrei* pre- and post-4.1 ka BP shows that the +0.08‰ shift in $\delta^{18}O$ is statistically significant (t value = 6.2, p < 0.01, n = 132), along with the +0.07‰ shift in mean $\delta^{13}C$ (t value = 3.3, p < 0.01, n = 132). This proxy, which we relate to winter mixing and IWM, shows a clear step change at 4.1 ka BP. The t-test for *G. sacculifer* also shows that the +0.08‰ shift in mean $\delta^{18}O$ values is statistically significant (t value = 3.8, p < 0.01, n = 128). Although these changes in mean $\delta^{18}O$ are small and on the order of the reproducibility of individual $\delta^{18}O$ measurements, they are significant when the variance of the population consisting of 60+ samples before and after 4.1 ka BP is considered. The shifts in mean $\delta^{18}O$ are also visually obvious in the records.

In addition, the SiZer analysis (Figure 4) objectively shows increases and decreases in the data that are significant (Chaudhuri and Marron, 1999), and both *G. sacculifer* and *N. dutertrei* exhibit significant increases at 4.1 ka BP for all smoothing bandwidths. We acknowledge that the t-tests for mean $\delta^{18}O$ of *G. ruber* pre- and post-4.1 ka BP are not significant (the 0.04-0.07‰ numbers referred to in this comment). We are not arguing for a stepped change in *G. ruber* $\delta^{18}O$ at 4.1 ka BP, but rather a period of increased values between 4.8 and 3.9 ka BP. The new $\delta^{18}O$ record of *G. ruber (*400-500µm) shows a double-peak maximum occurring at 4.1 and 3.95 ka BP that is related to seven discrete measurements with high $\delta^{18}O$ values (see Figure 1 below). These maxima are offset from the average $\delta^{18}O$ value by +0.18‰ (smoothed average), or up to +0.38‰ when considering the maximum individual measurement at 4.1 ka BP. The offsets from the average values exceed one standard deviation of the entire record from 5.4-3.0 ka BP, which is 0.13‰.

Despite the low signal to noise ratio of the *G. ruber* records, the long-term trends for both size fractions of *G. ruber* are similar. In fact, compared to the previously published $\delta^{18}O$ of *G. ruber* (315-400µm), the larger size fraction makes an even stronger case for the increased $\delta^{18}O$ spanning ~4.8-3.9 ka BP with a strong peak at 4.1 ka BP exceeding 1SD of the record, which is also apparent in the SiZer analysis.

[Figure]

**Figure 1.** Top: *G. ruber* (400-500μm) $\delta^{18}O$ shown with ±1SD, with three points around 4.1 ka BP circled. Bottom: $\Delta\delta^{18}O_{s-r}$ shown with ±1SD.

*Are the proxies chosen by Giesche et al. appropriate in these conditions to the task of reconstructing the summer and winter monsoons? We argue that the authors do not make a convincing case for this. First, their indicator for winter mixing, N. dutertrei, does not preferentially live in winter. Assuming that the limited sediment trap data cited by the authors is correct, the summer peak abundance in N. dutertrei is as important quantitatively as the winter peak due to its more extended temporal range (4 months compared to 1-2 months in winter). Thus isotopic signals in this species will be a mixed summer-winter 18O and not appropriate for detecting a winter monsoon signal.*

The available foraminifer trap data (Curry et al., 1992; Zaric, 2005) is limited in both temporal (1986-87) and spatial extent (1000 km SW of our coring location). We show the trap data as overlapping peaks in Figure 2: there are 2 traps (shallow and deep), and the deep trap has the longest time series (660 days). Unfortunately, the data collection of the deep trap stops just before the second winter season (end of October). This means that our summary figure shows foraminifera counts over 1 winter (2 traps) and 2 summers (2 traps). If we only compared the total sum of *N. dutertrei* from the shallow trap over 1 year, we would see 38% of total numbers stemming from summer months (JJA) and 62% in winter months (DJF) – indeed, the growth of this species is not restricted to one season. However, this would effectively only dampen the temperature signal recorded by *N. dutertrei* during

winter mixing. Furthermore, the temperature signal from winter mixing likely persists well beyond the winter months (Deser et al., 2003; Hanawa and Sugimoto, 2004), and therefore also affects the $\delta^{18}$O of thermocline-dwelling species throughout the summer. The exceptionally low $\delta^{18}$O values of *N. dutertrei* around 4.3 ka BP are best explained by warmer surface waters reaching deeper in the thermocline, and the differences between *N. dutertrei* and both *G. sacculifer* and *G. ruber* ($\Delta\delta^{18}O_{d-s}$, $\Delta\delta^{18}O_{d-r}$) suggest that cooler water may also be reaching the surface. Therefore, despite the scarcity of foraminifer trap data from our study area, we believe that our knowledge about winter mixing and foraminifer depth habitats provides sufficient information to interpret the $\delta^{18}$O signal of *N. dutertrei* in relation to winter mixing.

*Furthermore, interpretation of 13C values in this and other planktonic species is too simplistic given their known problems (e.g., possible shift in habitat, vital effects). Such problems are not discussed in the paper and interpretation is not even supported in the only cited reference (encyclopedia entry by Lynch-Stieglitz, 2006).*

Admittedly, the $\delta^{13}$C values of planktonic foraminifera are difficult to interpret. The basic principle is that surface waters of the ocean will have higher $\delta^{13}$C than deeper water due to uptake of more $^{12}$C by photosynthesis (Ravelo and Hillaire-Marcel, 2007), however, surface productivity is also increased by the upwelling of nutrient-rich bottom waters. With this in mind, lower $\delta^{13}$C values in the thermocline at 4.3 ka BP could reflect increased presence of deeper water (Sautter and Thunell, 1991), or possibly a decrease in productivity. We are not confident about the interpretation of the $\delta^{13}$C values, but its correlation to the $\delta^{18}$O signal of *N. dutertrei* warrants mention.

*In these conditions it is not productive to extend further our analysis of the paper as all interpretation and conclusions are vitiated by inappropriate basic assumptions. We urge the authors to consider a more conservative approach in interpreting this new data. It is evident to us that solving the salinity signal using forams in this region needs a more sophisticated approach (e.g., Ba/Ca in planktonics; temperature correction from Mg/Ca measurements, etc.). The alkenone-based SST estimates from Doose-Rolinski et al. can only be used to understand a qualitative indicative range of cooling as we now know that the high temperature plateau of the alkenone method limits its usefulness given the high SSTs in the region.*

Other methods such as Ba/Ca and Mg/Ca were explored on a few samples of *G. ruber* (315-400μm). Preliminary results show that Ba/Ca supports the $\Delta\delta^{18}O_{d-r}$ used to infer Indus River discharge, with very low Ba/Ca around 4.2 ka BP suggesting reduced freshwater discharge (see Figure 2 below) (Bahr et al., 2013). The Mg/Ca measurements lack a data point around 4.1 ka BP, but overall temperatures appear to be increasing between 4.2 and 3.8 ka BP. Additionally, using $\Delta\delta^{18}O_{d-r}$ would reduce the influence of a temperature signal on our proxies. It is premature to include these scarce measurements in this manuscript but we intend to develop these records further in the near future.

Evidence supports the basic assumptions made in the interpretation of the data, and statistics have demonstrated that these changes are significant. We have phrased our interpretations carefully to reflect uncertainities where they exist and have taken a conservative approach in interpreting the data. The signals we discuss exceed the 1SD variability of the data from 5.4-3.0 ka BP, and the technique of differencing $\delta^{18}$O minimizes

influence of other factors. We reject the claim that our study (and that of Staubwasser et al., 2003) is built on inappropriate basic assumptions and stand by our interpretation of the $\delta^{18}O$ and $\Delta\delta^{18}O$ signals in core 63KA.

[Figure]

**Figure 2.** Preliminary results from Mg/Ca and Ba/Ca measurements on *G. ruber* (315-400µm) indicate increasing temperatures over 4.2-3.8 ka BP, as well as lower river input around 4.2 ka BP, supporting the interpretations of Indus River discharge inferred from $\Delta\delta^{18}O_{d-r}$.

References mentioned in this response:

Bahr, A. Schönfeld, J., Hoffmann, J., Voigt, S., Aurahs, R., Kucera, M., Flögel, S., Jentzen, A., and Gerdes, A.: Comparison of Ba/Ca and as freshwater proxies: A multi-species core-top study on planktonic foraminifera from the vicinity of the Orinoco River mouth, Earth and Planetary Science Letters, 383, 45-57, 2013.

Chaudhuri, P., and Marron, J. S.: SiZer for exploration of structures in curves, Journal of the American Statistical Association, 94, 807-823, 1999.

Cullen, H. M., deMenocal, P. B., Hemming, S., Hemming, G., Brown, F. H., Guilderson, T., and Sirocko, F.: Climate change and the collapse of the Akkadian empire: Evidence from the deep sea, Geology, 28, 379-382, 2000.

Curry, W. B., Ostermann, D. R., Guptha, M. V. S., and Ittekkot, V.: Foraminiferal production and monsoonal upwelling in the Arabian Sea: evidence from sediment traps, Geological Society, London, Special Publications, 64, 93-106, 1992.

Deser, C., Alexander, M. A., and Timlin, M. S.: Understanding the persistence of sea surface temperature anomalies in midlatitudes, Journal of Climate, 16, 57-72, 2003.

Doose-Rolinski, H., Rogalla, U., Scheeder, G., Lückge, A., and Rad, U.: High-resolution temperature and evaporation changes during the late Holocene in the northeastern Arabian Sea, Paleoceanography and Paleoclimatology, 16, 358-367, 2001.

Hanawa, K., and Sugimoto, S.: 'Reemergence' areas of winter sea surface temperature anomalies in the world's oceans, Geophysical Research Letters, 31, 2004.

Karim, A., and Veizer, J.: Water balance of the Indus River Basin and moisture source in the Karakoram and western Himalayas: Implications from hydrogen and oxygen isotopes in river water, Journal of Geophysical Research: Atmospheres, 107, ACH-9, 2002.

Kumar, S. P., and Prasad, T. G.: Formation and spreading of Arabian Sea high-salinity water mass, Journal of Geophysical Research: Oceans, 104, 1455-1464, 1999.

Lambs, L., Balakrishna, K., Brunet, F., and Probst, J. L.: Oxygen and hydrogen isotopic composition of major Indian rivers: a first global assessment, Hydrological Processes, 19, 3345-3355, 2005.

LeGrande, A. N., and Schmidt, G. A.: Global gridded data set of the oxygen isotopic composition in seawater, Geophysical Research Letters, 33, 2006.

Milliman, J. D., Quraishee, G. S., and Beg, M. A. A.: Sediment discharge from the Indus River to the ocean: past, present and future, Marine Geology and Oceanography of Arabian Sea and Coastal Pakistan, 65-70, 1984.

Ravelo, A. C., and Hillaire-Marcel, C.: Chapter Eighteen the use of oxygen and carbon isotopes of foraminifera in Paleoceanography, Developments in Marine Geology, 1, 735-764, 2007.

Sautter, L. R., and Thunell, R. C.: Seasonal variability in the $\delta^{18}$O and $\delta^{13}$C of planktonic foraminifera from an upwelling environment: sediment trap results from the San Pedro Basin, Southern California Bight. Paleoceanography, 6, 307-334, 1991.

Staubwasser, M., Sirocko, F., Grootes, P. M., and Segl, M.: Climate change at the 4.2 ka BP termination of the Indus valley civilization and Holocene south Asian monsoon variability, Geophysical Research Letters, 30, 2003.

Staubwasser, M.: Late Holocene Drought Pattern Over West Asia. Climates, Landscapes, and Civilizations, 89-96, 2012.

Zaric, S.: Planktic foraminiferal flux of sediment trap EAST-86/90_trap, PANGAEA, doi: 10.1594/PANGAEA.264508, 2005.